# Leveraging multi-model season-ahead streamflow forecasts to trigger advanced flood preparedness in Peru

Colin Keating[1,2], Donghoon Lee[1,3], Juan Bazo[4,5], Paul Block[1]

[1]Department of Civil and Environmental Engineering, University of Wisconsin-Madison, Madison, USA
[2]Nelson Institute for Environmental Studies, University of Wisconsin-Madison, Madison, USA
[3]Climate Hazards Center, Department of Geography, University of California, Santa Barbara, USA
[4]Red Cross Red Crescent Climate Centre, The Hague, 2521 CV, the Netherlands
[5]Universidad Tecnológica del Perú (UTP), Lima, Perú

*Correspondence to*: Colin Keating (ckeating2@wisc.edu)

**Abstract.** Disaster planning has historically allocated minimal effort and finances toward advanced preparedness; however, evidence supports reduced vulnerability to flood events, saving lives and money, through appropriate early actions. Among other requirements, effective early action systems necessitate the availability of high-quality forecasts to inform decision making. In this study, we evaluate the ability of statistical and physically based season-ahead prediction models to appropriately trigger flood early preparedness actions based on a 75% or greater probability of surpassing the 80th percentile
of historical seasonal streamflow for the flood-prone Marañón River and Piura River in Peru. The statistical prediction model, developed in this work, leverages the asymmetric relationship between seasonal streamflow and the ENSO phenomenon. Additionally, a multi-model (least squares combination) is also evaluated against current operational practices. The statistical prediction demonstrates superior performance compared to the physically based model for the Marañón River by correctly triggering preparedness actions in three out of four historical occasions, while both the statistical and multi-
model predictions capture all four historical events when the required threshold exceedance probability is reduced to 50%, with only one false alarm. For the Piura River, the statistical model proves superior to all other approaches, correctly triggering 28% more often in the hindcast period. Continued efforts should focus on applying this season-ahead prediction framework to additional flood-prone locations where early actions may be warranted and current forecast capacity is limited.

## 1 Introduction and motivation

Globally, flood catastrophes lead all natural hazards in terms of mortality and cause billions of dollars in damages annually (Doocy et al., 2013; IFRC, 2020; Lee et al., 2018; Munich RE, 2012, 2018). Government agencies and relief organizations have historically prioritized disaster relief, allocating the majority of financial resources to response efforts in a reactionary mode, in lieu of pre-disaster preparedness (Perez et al., 2016). However, forecast based early action (FbA) initiatives are now recognized as a critical component of disaster risk reduction (IFRC, 2009). While no strict definition for FbA exists, the
term generally refers to initiatives that provide assistance and allocation of resources for preparation in advance of disasters

based on hydro-climate forecasts (Wilkinson et al., 2018). Empirical evidence demonstrates that actions taken in advance of a disaster can reduce loss of life and result in cost savings for relief organizations (Aguirre et al., 2019; Braman et al., 2013; Golnaraghi, 2012; Gros et al., 2019).

Forecast performance, uncertainty and hazard type contribute to the range and extent of potential early actions available. In 2013, a near-certain forecast prompted the evacuation of approximately 400,000 people in advance of Cyclone Phailin in India given a lead time of just four days (Harriman, 2014). While longer lead times allow for a greater range of potential early actions (Bazo et al., 2019), this must be balanced against corresponding increases in forecast uncertainty. To address this tradeoff, disaster managers seek low-regret actions, potentially in combination with a mechanism to halt early actions if

the threat of a disaster sufficiently drops, and thus avoid unnecessary costs (Wilkinson et al., 2018). While FbA was initially applied to acute and slowly evolving threats like tropical cyclones, more recent efforts have targeted hydrological threats including extreme rainfall and flooding (e.g., Gros et al., 2019). For example, in West Africa in 2008, preparatory actions, including prepositioning relief supplies and volunteer training, initiated based on a season-ahead forecast of above-average rainfall and high likelihood of floods, resulted in fewer deaths and lower response costs compared to previous flood events

when no early action was taken (Braman et al., 2013).

The question of when to initiate FbA requires integrating a hazard forecast with vulnerability and exposure information to estimate the impact of an extreme event. One commonly used method to trigger early action is to define a forecast threshold above which impacts are likely to occur based on historical data (Wilkinson et al., 2018). In London, actions to reduce

vulnerability for high-risk groups, such as ensuring indoor temperatures are below 26°C, are triggered when a forecast indicates temperatures of at least 32°C during the day and at least 18°C at night (Public Health London, 2018). This method accounts for the probabilistic nature of forecasts by requiring a predetermined level of forecast confidence; in London, a 60% probability of reaching the temperature thresholds is required.

**Table 1:** Contingency table demonstrating potential outcomes of forecast based action.

|  | Extreme Event | No Extreme Event |
|---|---|---|
| **Early Action** | *Worthy action* | *Action in vain* |
| **No Early Action** | *Failure to act* | *Worthy inaction* |

Note: Adapted from Lopez et al. (2017), Table 1.

When linking early action based on probabilistic forecasts to the occurrence of extreme events, four scenarios are possible (Table 1) where worthy action and worthy inaction are preferred. The risk of acting in vain, when early action is initiated but an extreme event fails to materialize (Lopez et al., 2017), is often viewed as a major barrier to scaling up FbA (Tanner et al.,

2019). However, studies have found that, when compared to a late response, early action is almost invariably cheaper: a late response can be two to six times more costly than actions in vain (Cabot Venton et al., 2012). Additionally, financial based

actions such as unconditional cash disbursements targeting vulnerable households can yield a benefit regardless of whether or not the event occurs (Wilkinson et al., 2018). Forecast models that proficiently predict extreme events at lead times permitting early action are critical for minimizing false positives and false negatives. In addition to short term weather

forecasts which are commonly viewed as skillful, medium to long range climate forecasts have also been demonstrated to improve preparedness protocols, resulting in reduced mortality, morbidity, and resource demands (Braman et al., 2013), yet their applications have been limited predominantly as a result of moderate forecast performance and significant uncertainty.

Improvement in the skill of hydrologic models over the last several decades has aided the development of FbA systems for

flooding. Among hydrologic models, those that are physically based (or dynamical) simulate physical processes such as infiltration and runoff to produce streamflow predictions and are often forced with climate predictions downscaled from general circulation models (GCMs) or numerical weather models. Statistical (also called empirical or data-driven) models forgo the parameterization of complex physical processes in favor of understanding the lagged relationships between precipitation or streamflow and antecedent land, atmosphere and ocean conditions. Statistical and physical models have been

successfully applied to seasonal prediction of hydrologic variables including precipitation and streamflow (e.g., Badr, et al., 2013; Block & Rajagopalan, 2009). Both frameworks have their own set of advantages and disadvantages with prediction skill varying according to season and location (Infanti and Kirtman, 2014). While statistical models are not intended to provide a complete understanding of the hydro-climate system, they offer an appealing complement to physically based models by focusing solely on the prediction variable of interest (Zimmerman et al., 2016).


A common traditional approach for statistical hydrologic modeling is multiple linear regression (MLR), which relates a predictand to the linear combination of several predictor variables (Moradkhani and Meier, 2010). For categorical streamflow forecasts, logistic regression (for two categories) or polytomous logistic regression (for three or more categories) has been used successfully (e.g., Wei and Watkins, 2011). Because these methods are prone to multicollinearity due to the

overlapping signals present in many hydroclimate variables, techniques such as principal component regression (PCR; a combination of principal component analysis and MLR) and partial least squares regression (e.g., Lala et al., 2020) are employed to address this challenge. More recently, machine learning techniques, adept at capturing nonlinear relationships between predictors and a predictand, have been successfully applied to hydroclimate forecasting, including artificial neural networks (Zealand et al., 1999), random forest classification (Ali et al., 2020; Lala et al., 2020) and support-vector machines

(Asefa et al., 2006; Shabri and Suhartono, 2012). There is also increasing recognition that hybrid approaches combining statistical and dynamical techniques can offer greater accuracy than even state-of-the-art dynamical models (Cohen et al., 2019).

Multi-model techniques have been developed based on the assumption that errors present in individual models may cancel

out, thus providing a multi-model average with greater skill than any individual model, and to bound forecast uncertainty

based on the spread of model predictions. Several methods of combining models include equal weighting, linear regression and Bayesian methods that assign weights according to the probability that the model in question has the highest skill (e.g., Gneiting and Raftery, 2005). In some cases, multi-model ensembles have been shown to significantly increase forecast skill over the best performing individual model (e.g., Regonda et al., 2006), while not in other cases. For example, Bohn et al. (2010) note only modest improvement when using a least-squares weighted multi-model.

This study evaluates multiple season-ahead forecast approaches, namely locally tailored statistical and existing global-scale physical models, to individually and collectively inform advanced flood preparedness actions, using Peru as a case study. Typically, only physically based forecast approaches are used operationally, however augmenting with a locally tailored statistical forecast may considerably improve forecast performance and opportunities for preparedness. In this paper, we use the term "season-ahead prediction" to describe forecasting the mean streamflow for an upcoming three-month season issued at the start of that season. Ideally, a season-ahead prediction of January-February-March streamflow would be issued on December 31st and represents a prediction of the average streamflow over the upcoming three months. In practice, due to lags in data availability and for purposes of direct comparison with a physically-based model, forecasts developed in this study are issued on the 10th day into the three-month season.

## 2 Case study in Peru

### 2.1 Flood impacts in Peru

Peru experiences catastrophic flooding with relative frequency, resulting in significant adverse economic and health impacts. In northwest Peru, flooding caused by extreme rainfall during El Niño events in 1982-83, 1997-98 and the 2017 "coastal El Niño" each incurred damages exceeding USD$5 billion (in 2020 dollars) and collectively resulted in over 1000 deaths (French & Mechler, 2017; Venkateswaran et al., 2017). Flooding in the Peruvian Amazon basin affected over 300,000 people in 2012 (IFRC, 2012) and over 100,000 people in 2015 (IFRC, 2015). Floods prevent access to safe drinking water, disrupt livelihoods centered around farming and fishing, and can force residents to relocate from low-lying areas (IFRC, 2019). Health impacts of extreme flooding include increased incidence of acute diarrheal disease, arboviral diseases, malaria, and water-borne diseases (Caviedes, 1984; IFRC, 2019).

### 2.2 Hydroclimatology of Peru

While floods are common throughout many regions of Peru, climate and hydrology vary dramatically. The hydroclimatology of Peru is broadly characterized by a disruption of tropospheric flow caused by the Andes cordillera, which maintains an arid climate along the Pacific coast and wet conditions in the Amazon basin to the east (Garreaud et al., 2009). Particularly along coastal Peru, a major source of interannual variability in precipitation and temperature is controlled by the El Niño Southern

Oscillation (ENSO) phenomenon, a system of ocean-atmosphere feedbacks in the tropical Pacific (Garreaud et al., 2009). In the southern coastal region, the warm, positive phase of ENSO (El Niño) is associated with below average precipitation (Wu et al., 2018). In northwest Peru, strong El Niño years are often associated with above average precipitation, most notably during the 1982-83 and 1997-98 El Niño events which coincided with extreme rainfall and flooding (Bayer et al., 2014).

However, the impacts of similarly intense El Niño events are variable. Despite very strong El Niño conditions in 2015-2016, rainfall and flood impacts in Peru were minimal (French and Mechler, 2017; Ramirez and Briones, 2017; Venkateswaran et al., 2017). El Niño events can span the equatorial Pacific region (e.g., 1982-83, 1997-98) or they can be confined to the coast of northern Peru and Ecuador (Ramirez and Briones, 2017). The latter type is known as a "coastal El Niño" or "El Niño costero" and has occurred in 1925 and 2017, in both cases resulting in extreme rainfall and flooding (Ramirez and Briones, 

2017; Takahashi and Martínez, 2017). While El Niño conditions are associated with extreme events along the coast, La Niña (cool, negative phase of ENSO) conditions can also produce slightly higher than average streamflow (Figure 2b).

In the Amazon basin, the influence of climate variables on flood risk remains understudied (Towner et al., 2020) as a result of the nonlinear relationship between precipitation and streamflow (Stephens et al., 2015). Hydrometeorological regimes in 

the Amazon basin are diverse and are driven by seasonal warming of the northern and southern hemispheres and the migration of the Intertropical Convergence Zone (Espinoza Villar et al., 2009). Precipitation in the Peruvian austral summer (DJFM) is dominated by the South American Monsoon season which enhances the north Atlantic trade wind (Zhou and Lau, 1998) as well as by deep convection that recycles moisture over Amazonia (Garreaud et al., 2009). El Niño conditions and above-average sea surface temperatures (SST) in the tropical north Atlantic, south Atlantic, and Indian Oceans are associated 

with decreased rainfall in the northern portion of the basin and increased rainfall in the south (Marengo, 2004). La Niña conditions are weakly associated with increased precipitation in the western Amazon basin (Garreaud et al., 2009).

### 2.3 Flood early action plan

In October 2019, the International Federation of Red Cross and Red Crescent Societies (IFRC) approved an Early Action Plan (EAP) submitted by the Peruvian Red Cross for flooding in the Peruvian Amazon. The plan is based in part on an 

extension of the Global Flood Awareness System (GloFAS) called GloFAS-seasonal, a global streamflow forecast model developed by the European Centre for Medium-Range Weather Forecasts (ECMWF) that couples seasonal climate forecasts from GCMs to a physically based hydrology model (Emerton et al., 2018). Early actions, which involve the prepositioning of supplies and release of funds, are triggered when 75% of GloFAS ensemble members forecast streamflow above the 80[th] percentile (IFRC, 2019) at a 45-day lead time. Because GloFAS exhibits only modest forecast skill in Peru when detecting 

floods at short lead times (Bischiniotis et al., 2019), there is an opportunity to leverage complementary prediction frameworks to improve forecast performance. Similarly, an EAP is in development for the Piura basin in coastal northwest Peru to address extreme precipitation and flooding.

**2.4 Case study locations**

Study locations prone to riverine flooding were identified by collaborators at the Red Cross Climate Center in Lima, Peru,

and the EAPs, namely the Marañón River at San Regis and the Piura River at Puente Sánchez Cerro (Figure 1). The Marañón is a tributary to the Amazon River, east of the Andes, with a basin covering approximately one-half (362,000 km$^2$) of the Peruvian Amazon River basin. Here, tropical lowland forest (below 600 m elevation) is the dominant ecozone followed by tropical montane forest (above 600 m elevation) (Kvist and Nebel, 2001). The Piura River basin above Puente

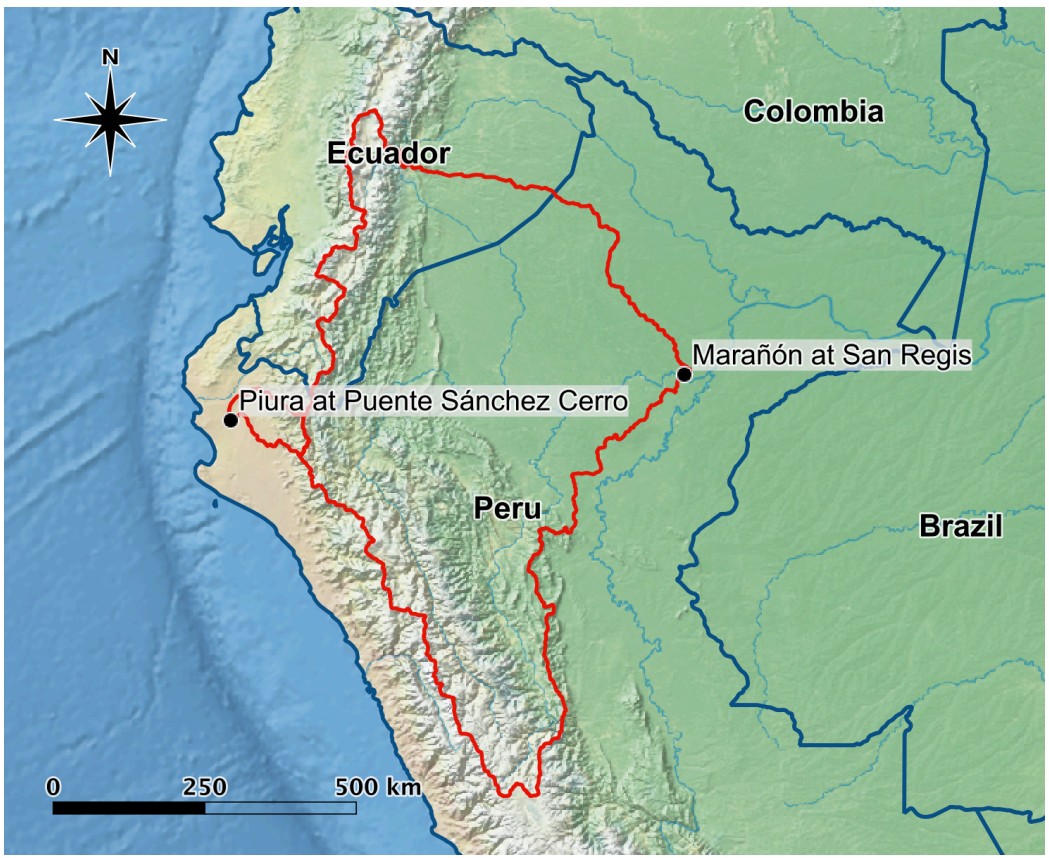

**Figure 1:** Case study locations with catchment boundaries delimited in red. Shading represents idealized land cover. Made with Natural Earth (naturalearthdata.com).

Sánchez Cerro is significantly smaller in size (7,435 km$^2$), consists of tropical shrubland and tropical mountain systems and is generally classified as arid with precipitation averaging less than 50 mm/year for elevations below 500 m (FAO 2001; Rodriguez et al., 2005). Throughout this paper, the names of the monitoring stations will be used to describe the stations and

the basins they delimit.

## 2.5 Streamflow variability

Daily streamflow data for each location (1999-2017 at San Regis, 1971-2017 at Puente Sánchez Cerro) was provided by the Peruvian Meteorological Agency, El Servicio Nacional de Meteorología e Hidrología del Perú (SENAMHI), who performed appropriate quality assurance. Monthly mean streamflow at Marañón exhibits a sinusoidal autocorrelation structure, with statistically significant autocorrelation at one- and two-month lags as well as at interannual timescales. In contrast, streamflow at Piura exhibits significant autocorrelation at up to a three month lag yet minimal autocorrelation at interannual timescales, indicating a greater degree of variability in successive years. This is predominantly an effect of catchment size and watershed memory, and an important feature for streamflow prediction.

The high flow season during which floods are likely to occur is computed using an approach modified from Lee et al. (2015) and is defined as the three consecutive months with the largest combined number of days with streamflow values in the top 1% of all days in the historical record. For Marañón, the high flow season is March, April and May (MAM); for Piura, it is February, March and April (FMA). Testing this approach with a slightly lower threshold to define high flow days (3% and 5%) returns the same high flow season, further validating the seasons selected. The high flow season for Marañón identified via this methodology is similar to the IFRC's characterization of flood season in the Amazon basin as running from December to April (IFRC, 2019). At Marañón, all daily observations in the top 1% occurred in MAM and the annual maximum occurred in MAM in 17 out of 19 years; at Piura, 87% of daily observations in the top 1% occurred in FMA while the annual maximum discharge occurred in FMA in 40 out of 47 years. Clearly, high flow conditions occur outside these seasons, however in this study these will not be captured as the focus is on the likelihood of high flow conditions within the target season only.

## 3 Statistical approach to season-ahead streamflow prediction

### 3.1 Potential local-scale predictor variables

Ocean-land-atmospheric variables representative of slowly evolving hydro-climatic conditions offer prospects for predicting streamflow from a season-ahead lead. This includes considering pre-season large-scale ocean-atmosphere teleconnections and basin-scale hydrologic processes such as observed streamflow, precipitation, soil moisture, and temperature (Table 2). Predictions of seasonal (three month) average streamflow (m³/s) are issued on the 10th day into the three-month high flow season identified in Sect. 2, leveraging predictors based on values in the preceding months. Practically, issuing the forecast ten days into the forecast season allows time for large-scale climate data to be made available online, while also fostering a more direct comparison with GloFAS as described in Sect. 3.4.

Precipitation data used in this study leverages the Peruvian Interpolation data of SENAMHI's Climatological and hydrological Observations (PISCO) v2.1 dataset (Aybar et al., 2020), provided by SENAMHI and accessed via the International Research Institute for Climate and Society (IRI; http://iridl.ldeo.columbia.edu). PISCO contains monthly and daily precipitation at a 0.1 degree grid resolution from 1981 to 2017, and is based on the Climate Hazards group InfraRed Precipitation with Stations (CHIRPS; Funk et al., 2015) quasi-global precipitation product calibrated with SENAHMI station data. Basin-averaged precipitation over January-February is included as a potential predictor for the Marañón at San Regis (Table 2). January and February precipitation each also correlate significantly with streamflow, though less so compared to the January-February average; to maintain model parsimony we included only the latter as a potential predictor. The Piura catchment is approximately 2% the size of the Marañón and only basin-averaged precipitation in January significantly correlates with streamflow (Table 2).

Soil moisture data (0.5°, monthly) is provided by the National Oceanic and Atmospheric Administration (NOAA) Climate Prediction Center (Fan and van den Dool, 2004). Atmospheric moisture transport can occur over long distances and across catchment boundaries; to capture potential signals of soil moisture on streamflow variability, a principal component analysis is conducted on one-month ahead gridded soil moisture across northern South America, and the first principal component (PC) is retained as a potential predictor. Basin-averaged mean air temperature in the month prior to the forecast, provided by NOAA (https://psl.noaa.gov/) is also considered (Table 2).

**Table 2:** The suite of potential predictor variables for the statistical forecast model and their Pearson correlation coefficient with FMA streamflow at Piura at Puente Sánchez Cerro and MAM streamflow at Marañón at San Regis; * indicates statistically significant correlations ($p < 0.05$). SST and SLP predictor spatial extents are determined by NIPA (Figure 3) and correlations are presented by phase. J (F) indicates January (February).

| Potential Predictor | Abbreviation | Spatial Region | Time Frame Piura | Time Frame Marañón | Pearson Correlation with Streamflow Piura | | | Pearson Correlation with Streamflow Marañón | |
|---|---|---|---|---|---|---|---|---|---|
| Streamflow | SF | - | J | F | 0.84* | | | 0.84* | |
| Precipitation | P | Basin-Avg | J | JF | 0.88* | | | 0.68* | |
| Soil Moisture | SM | 1st PC of statistically significant ($p < 0.05$) regions within 12N to 23S, 35W to 81.5W | J | F | 0.69* | | | 0.74* | |
| Air Temperature | T | Basin-Avg | J | F | 0.26 | | | 0.11 | |
| GCM Precipitation Forecast | P(GCM) | 4.5S to 5.5S, 79.5W to 80.5W | FMA | - | 0.84* | | | - | |
| | | | | | El Niño | Neutral | La Niña | El Niño | La Niña |
| Sea Surface Temperature | SST | 1st PC of NIPA-identified regions | NDJ | DJF | -0.79* | -0.90* | 0.85* | -0.93* | -0.80* |
| Sea Level Pressure | SLP | 1st PC of NIPA-identified regions | J | F | -0.82* | -0.74* | 0.79* | 0.90* | -0.72* |

Given that the Piura basin is relatively small and within-season precipitation is an important contributor to seasonal streamflow, FMA precipitation (mm/day) predictions derived from the mean of two GCM members (NASA GEOS-S2S and NCEP CFSv2) of the North American Multi-Model Ensemble (NMME) (Kirtman et al., 2014) are also evaluated. The two models have exhibited superior performance in terms of RMSE, temporal correlation, and Heidke Skill Score in northwest Peru compared to other NMME models when simulating January, February and March precipitation across lead times of one to six months (Wang et al., 2021). Individually, each model's FMA precipitation prediction correlates with streamflow at 0.76; when averaged, correlation increases to 0.84 (Table 2).

## 3.2 Potential large-scale predictor variables

A common approach for identifying SST regions for use as predictors is to search for stable correlations between the predictand (streamflow in this case) and SSTs over a moving window of historical data (Gámiz-Fortis et al., 2010; Ionita et al., 2015). However, the state of ENSO can influence the mean state of the atmospheric-oceanic system, which in turn affects

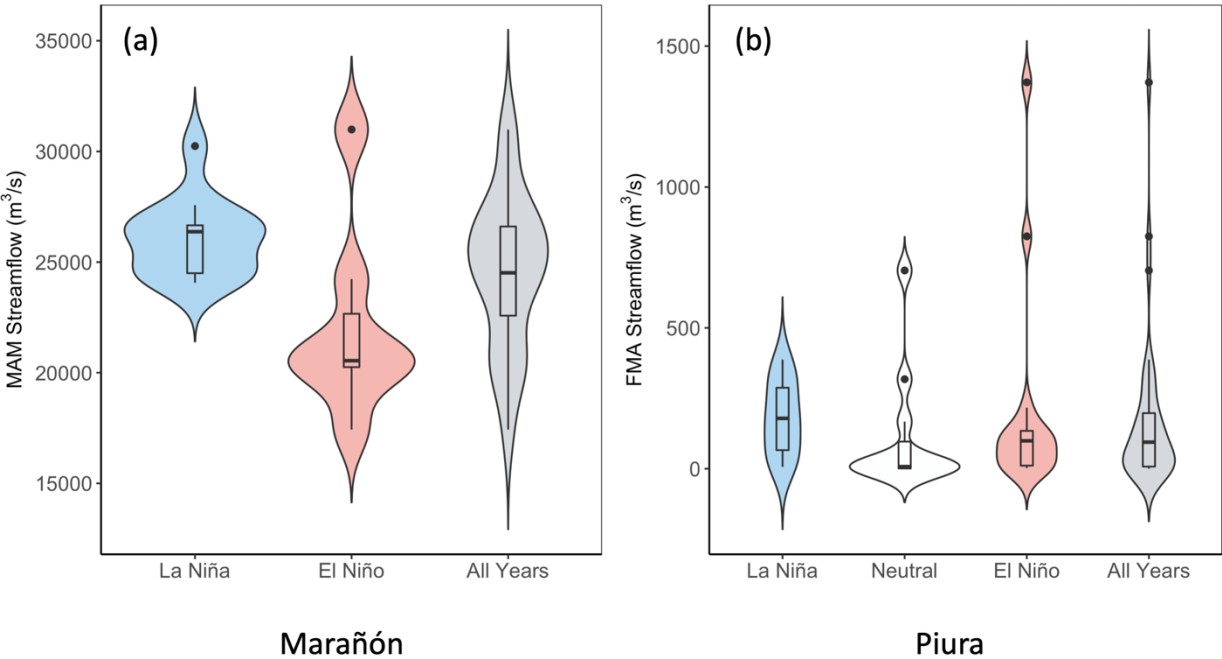

**Figure 2:** Violin plots of seasonal streamflow by ENSO phase. For the Marañón River at San Regis (n=19), twelve historical years are classified as La Niña conditions (MEI ≤ 0) and seven are classified as El Niño conditions (MEI > 0). For the Piura River at Puente Sánchez Cerro (n=36), eleven years are classified as La Niña (MEI ≤ -0.5), eleven as neutral (-0.5 < MEI < 0.5), and fourteen as El Niño conditions (MEI ≥ 0.5).

the relevant teleconnections between SSTs and precipitation or streamflow (Zimmerman et al., 2016). This asymmetric
relationship between ENSO and streamflow may prove challenging from a traditional modeling perspective. At our study
sites, the distributions of seasonal streamflow shift and change shape according to the state of ENSO, though significant
variability within each phase exists (Figure 2). A Nino Index Phase Analysis (NIPA; Giuliani et al., 2019; Zimmerman et al.,
2016) approach is advantageous in such cases, capturing the variance and signals within each phase separately, and thus
addressing the overall asymmetric challenges.


The approach proposed by Zimmerman et al. (2016) is adopted to select global SST and Sea Level Pressure (SLP) regions
exhibiting strong teleconnections with streamflow at our study sites. The selection of these regions is conditioned on the
preseason state of ENSO (NDJ for Piura and DJF for Marañón) as represented by the average Multivariate ENSO Index
(MEI) value (Wolter and Timlin, 2011). Historical years are categorized according to the preseason average value of MEI.
For this analysis, three categories are selected for Piura and two for Marañón (Figure 2). While including more bins may
potentially provide additional unique streamflow information by further distinguishing climate system states, this needs to be
balanced against available observational data. For Piura, the three categories are generally representative of El Niño, La Niña
or neutral conditions, per NOAA's definition (NOAA, 2020). The short historical dataset at Marañón at San Regis limits
categorizing into two phases delineated as positive and negative MEI values. (While a two-phase model for Piura was also
tested, the 3-phase model improves performance, including in years critical for disaster preparedness.) For years classified
within each phase, observed target season streamflow is correlated with global pre-season SSTs from the NOAA

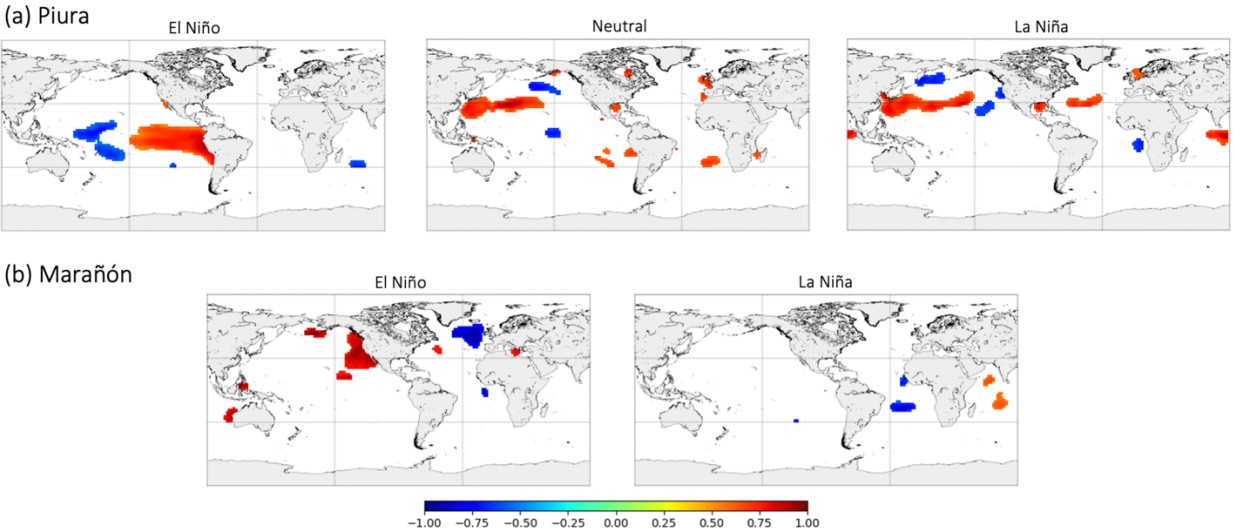

**Figure 3:** Correlation maps of seasonal streamflow at a) Piura (FMA) and b) Marañón (MAM) with pre-season SSTs by ENSO phase.
Only regions statistically significantly correlated at p<0.05 are included.

Extended Reconstructed Sea Surface Temperature V3b dataset (Smith et al., 2008), a global gridded dataset of monthly mean SSTs at a two-degree resolution from 1854 to present accessed via the IRI data library. Of the SST regions statistically significantly correlated with streamflow (Figure 3), the first and second PC is extracted as a potential predictor in the statistical model. For Piura (Marañón) the first and second PCs explain 83% and 7% (84% and 6%) of the variance respectively and only the first PC significantly correlates with streamflow. Selecting SST regions based on the preseason

state of the Niño 1+2 anomaly index instead of MEI did not materially change results at Piura.

Given that SLP evolves more quickly than SSTs, only the single month values prior to the target season are evaluated, otherwise the process mirrors SST selection. SLP data is from the NCEP/NCAR Climate Data Assimilation System I (Kalnay et al., 1996) and accessed via the IRI data library.

**3.3 Statistical prediction model**

A principal component regression (PCR; coupled principal component analysis and multiple linear regression) framework is adopted to predict seasonal (3-month) average seasonal streamflow derived from daily streamflow observations obtained from SENAMHI as described in Sect. 2.5. The forecast for each location is composed of sub-models (multiple linear regression) composed of years in a particular climate state, as represented by the preseason (3-month average) value of MEI.

This produces two sub-models for the Marañón River at San Regis and three for the Piura River at Puente Sánchez Cerro. A hindcast assessment is conducted by evaluating each year in the historical record using the appropriate sub-model to predict seasonal streamflow. For example, in 1998, the preseason (NDJ) average MEI value is 2.43, thus the positive phase sub-model is selected to predict Piura River FMA streamflow. Predictor variable types listed in Table 2 may be included in some sub-models and not others, subject to their correlation with streamflow in that phase (Table 3). To be included, the predictor

in question must be both significantly correlated with streamflow across all years and significantly correlated with streamflow in the subset of phase-specific years. A principal component analysis is conducted on eligible predictors which are first scaled to have a unit variance. A subset of PCs is retained according to North's Rule-of-Thumb (North et al., 1982) for input into the multiple linear regression, given as:

$$y_t = \beta_0 + \beta_1 x_{1,t} + \cdots + \beta_n x_{n,t} + e \, , \qquad\qquad\qquad (1)$$

where $y_t$ is observed seasonal streamflow in year $t$, $\beta_0$ is a constant, $\beta_1 \ldots \beta_n$ are regression coefficients, $x_{1,t} \ldots x_{n,t}$ are the PCs retained, and $e$ is the residual or error. If North's Rule-of-Thumb indicates that no PCs are non-overlapping then only the first PC is retained.

The creation of probabilistic forecasts are essential as early action decisions are conditioned on the forecast likelihood of an

extreme event exceeding the 80th percentile. For each sub-model, a drop-one-year cross validation hindcast is constructed, refitting the regression coefficients each year, to produce one deterministic seasonal streamflow prediction per year. When

model residuals are normally distributed, according to the Shapiro-Wilk test with alpha=0.05, an error distribution is created by taking 1000 random samples. Otherwise, an error distribution is derived by directly sampling the model residuals with replacement 1000 times. The resulting error distribution is then added to the cross-validated deterministic prediction to create a probabilistic prediction of average streamflow in the upcoming season. This process is repeated for each year to create a probabilistic hindcast for all years in the sub-model. Hindcasts from each sub-model are subsequently joined to create a full observational period probabilistic hindcast.

### 3.4 GloFAS and multi-model predictions

Monthly hindcasts over the period 1981-2017 from the physically based GloFAS Seasonal model (version 2.0) for the two study locations are available from ECMWF (https://www.globalfloods.eu/general-information/data-and-services/). Both study locations were used for model calibration (E. Zsoter, personal communication, May 6, 2021). GloFAS forecasts are initialized on the first day of every month and become publicly available on the 10th day of the month. They consist of 25 ensemble members predicting mean weekly streamflow to 17 weeks out; predictions for weeks 1-13 (approximately three months) are retained. A mean bias correction is applied to the GloFAS ensemble mean according to the difference between mean observed and predicted seasonal streamflow across all years. A quantile mapping approach, relating the cumulative distributions functions of observed and predicted streamflow, was also tested (Hashino et al., 2006); however, forecast skill did not substantially differ from the mean bias correction approach. In addition to evaluating the statistical model and GloFAS independently, a multi-model forecast is also constructed utilizing a least squares linear regression to assign weights according to the relative Pearson correlation strength between observed streamflow and each model's predictions (P. J. Block et al., 2009).

### 3.5 Forecast verification and performance measures

Forecast performance for the three models (statistical, GloFAS, and multi-model) is evaluated at both locations by Pearson correlation coefficient, Rank Probability Skill Score (RPSS), Probability of Detection (POD), False Alarm Ratio (FAR) and Threat Score (TS).

RPSS is an extension of the rank probability score (RPS), which measures the categorical accuracy of a forecast (Wilks, 2011). Here, two categories are selected to represent high flow and non-high flow conditions, with the 80th percentile of observed seasonal streamflow representing the threshold. The RPS is the sum of the squared differences between the forecast and observed categorical probabilities, and is given as:

$$\text{RPS} = \frac{1}{J-1}\sum_{m=1}^{J}\left[\left(\sum_{j=1}^{m}p_j\right)-\left(\sum_{j=1}^{m}o_j\right)\right]^2, \tag{3}$$

where $J$ is the number of categories, $y_j$ is the forecast probability in the $j$th category, and $o_j$ is 1 if the event is observed in that category, otherwise 0.  RPS scores range from 0 to 1. RPSS indicates the relative skill of the forecast compared to a reference forecast and takes the form:

$$\text{RPSS} = 1 - \frac{\text{RPS}}{\text{RPS}_{reference}} . \tag{4}$$

RPSS can vary from -∞ to 1; values above 0 are considered skillful compared to the reference forecast, and a value equal to 1 indicates a perfect categorical forecast. Mean RPSS values across all hindcast years are presented; the reference forecast is based on historical averages (i.e. climatology).

POD, or "hit rate," describes the fraction of observed extreme (e.g., high flow) events that are correctly predicted and is
calculated as:

$$\text{POD} = \frac{hits}{hits+misses}, \tag{5}$$

where a perfect score is 1 (Wilks, 2011). Because POD can be artificially improved by issuing more extreme predictions, it must be evaluated in combination with FAR. FAR describes the fraction of predicted extreme events that did not occur, or "false alarms", calculated as:

$$\text{FAR} = \frac{false\ alarms}{hits\ +false\ alarms}, \tag{6}$$

where a perfect score is 0 (Wilks, 2011).

TS, also called the "critical success index," is the number of predicted extreme events divided by the total number of times that an extreme event is either predicted or observed, calculated as:

$$\text{TS} = \frac{hits}{hits+misses+false\ alarms}, \tag{7}$$

where a perfect score is 1 (Wilks, 2011). TS is preferred over accuracy (the sum of true positives and true negatives divided by the total number of events) for situations where the extreme category is rarely observed. As previously stated, the extreme category is classified as seasonal streamflow values in the top 20% (80th percentile) of observations – four events for Marañón and seven events for Piura.

# 4 Results

## 4.1 Large-scale predictor regions

The locations of SST regions that correlate significantly with streamflow vary according to the phase of ENSO (Figure 3). Piura streamflow in El Niño years is positively associated with equatorial Pacific SSTs, encompassing the Niño 1+2 and Niño 3 regions (Figure 3a). This finding aligns with previous work demonstrating that above-average precipitation in northwest Peru is driven primarily by ENSO (e.g., Lagos et al., 2008). Strong El Niño years (e.g. 1983, 1998) have a tendency to lead to extreme flooding in northwest Peru, though floods have also affected the region in other ENSO phases, for example, in 2008, a moderate La Niña (EM-DAT, 1988). Piura streamflow variability in neutral and La Niña years is associated with SSTs in the northwest Pacific, north Atlantic, and tropical Indian Oceans (Figure 3a). This is similar to the findings of Bazo et al. (2013) who show an influence of SST anomalies in the tropical Indian and Atlantic Oceans (in addition to the tropical Pacific) on precipitation in northwest Peru.

Marañón streamflow during El Niño years is positively (negatively) associated with northeast Pacific (northwest Atlantic) SSTs (Figure 3b). In La Niña years, when average Marañón streamflow is greater and hydrologic disasters are more common in Amazonian Peru (Rodríguez-Morata et al., 2018), streamflow is associated with SST regions in the tropical Atlantic and Indian Oceans. While El Niño episodes have been linked to below-average precipitation in the Amazon basin (Garreaud et al., 2009; Marengo, 2004), significant teleconnections between equatorial Pacific SSTs and Marañón streamflow are not identified here (Figure 3b).

## 4.2 Final predictor selection

Of the potential predictors listed in Table 2, Table 3 shows the subset selected for each statistical forecast sub-model based on correlation significance as described in Sect. 3.3. The first PC of statistically significant pre-season SST regions is included in all sub-models for both locations. For Marañón's negative phase sub-model, no PCs are unique by North's Rule-

**Table 3:** Final predictors included in each sub-model.

| Site | Sub-model | Number of observations | Predictors retained from Table 2 | PCs retained | PC1 % variance explained | PC2 % variance explained |
|------|-----------|------------------------|----------------------------------|--------------|--------------------------|--------------------------|
| Marañón | Negative Phase | 12 | SST, SLP, SF, SM | 1 | 61 | 22 |
| | Positive Phase | 7 | SST, SLP, SF, SM, P | 1 | 87 | 9 |
| Piura | Negative Phase | 11 | SST, SLP, SM, P(GCM) | 1 | 74 | 15 |
| | Positive Phase | 14 | SST, SLP, SF, SM, P, P(GCM) | 1 | 78 | 13 |
| | Neutral Phase | 11 | SST, SLP, SM, P, P(GCM) | 1 | 68 | 15 |

of-Thumb; in all other cases only the first PC is unique. Pre-season streamflow is included in both sub-models for Marañón, in line with its greater temporal autocorrelation, while it is included in only the positive phase sub-model for Piura. No pre-season precipitation observations are included for Marañón; for Piura the GCM precipitation forecast is included in the negative phase sub-model and pre-season observed precipitation is included in the positive and neutral phase sub-models. For all sub-models the predictand is seasonal (3-month) average streamflow (m$^3$/s), which is predicted by (multiple) linear regression using the PC(s) retained in Table 3.

### 4.3 Statistical model forecasts

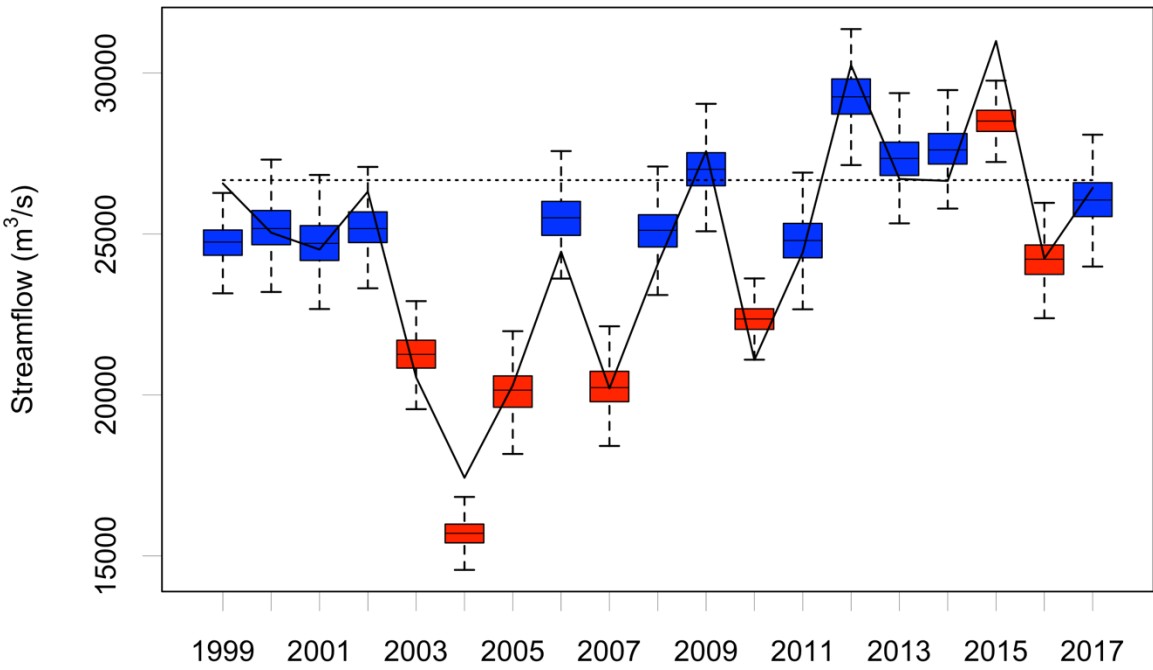

**Figure 4:** Marañón River at San Regis MAM streamflow hindcast using the statistical prediction model. The black solid line illustrates observed MAM streamflow; the black dotted line indicates the 80[th] percentile of MAM observed streamflow. Red (blue) boxes represent years with pre-season El Niño (La Niña) conditions.

The primary focus of this study is to predict the occurrence of high flow conditions to initiate flood preparedness actions, based on a sufficient percentage of the probabilistic prediction surpassing a pre-defined threshold. The probabilistic statistical forecast model at each location effectively captures interannual variability and extremes (Figs. 4 and 5). For the two most extreme years in the observed record (2012 and 2015 for Marañón; 1983 and 1998 for Piura), the full distribution of predicted streamflow falls above the 80th percentile of observed streamflow (black dashed line). In these years, decision-

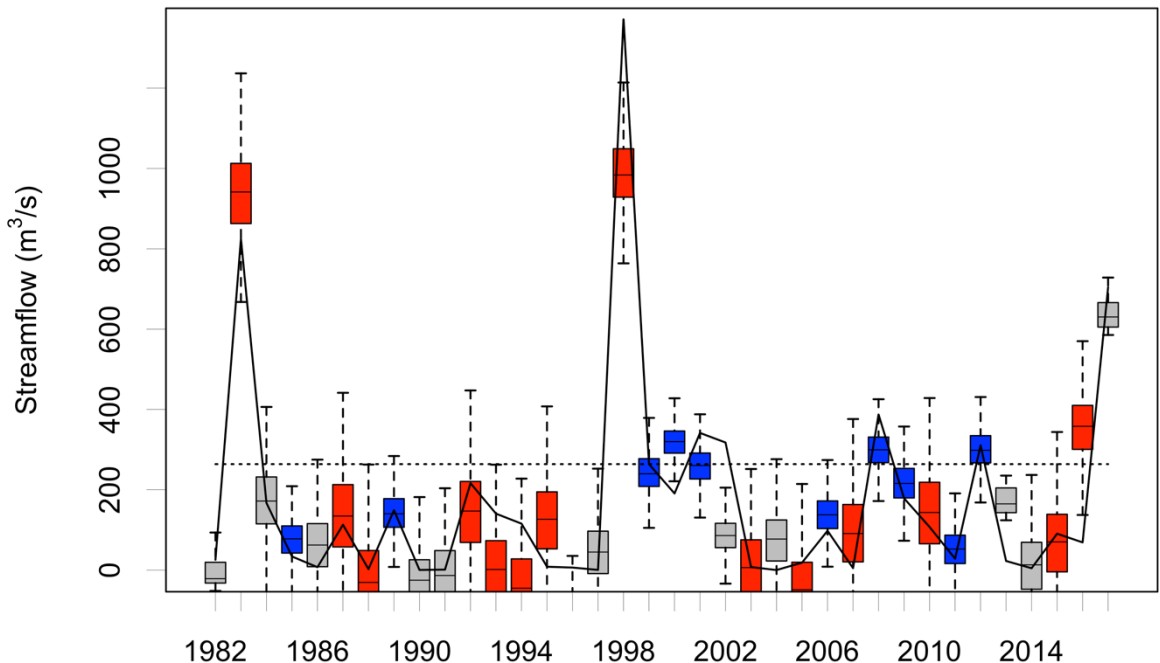

**Figure 5:** Piura River at Puente Sánchez Cerro FMA streamflow hindcast using the statistical prediction model. The black solid line illustrates observed FMA streamflow; the black dotted line indicates the 80th percentile of FMA observed streamflow. Red (blue) boxes represent years with pre-season El Niño (La Niña) conditions.

**Table 4:** Contingency table for statistical, GloFAS, and multi-model predictions of high flow (top 20%) and low flow (bottom 80%) MAM (FMA) streamflow for the Marañón (Piura) River.

| | | | Observed Conditions | | | | | |
|---|---|---|---|---|---|---|---|---|
| | | | Statistical | | GloFAS | | Multi-model | |
| | | | Low | High | Low | High | Low | High |
| **Predicted Conditions** | Marañón | Low | 14 | 0 | 13 | 2 | 14 | 0 |
| | | High | 1 | 4 | 2 | 2 | 1 | 4 |
| | Piura | Low | 26 | 3 | 27 | 5 | 28 | 4 |
| | | High | 2 | 5 | 1 | 3 | 0 | 4 |

makers are highly certain of an impending extreme event. However, for the majority of years, some smaller fraction of the forecast distribution falls above the 80th percentile threshold, presenting a greater challenge (less certainty) in decision making. When evaluated categorically, the Marañón forecast model identifies all four high flow years while the forecast for Piura identifies six out of eight (Table 4). El Niño years are associated with lower forecast uncertainty for Marañón; the

average standard deviation of error distributions is 20% smaller than in La Niña years. For Piura, La Niña conditions result in lower forecast uncertainy; the average standard deviation of error distributions is 58% larger for years in the neutral phase and 113% larger in El Niño years. Despite low streamflow in many years, the Piura forecast's mean prediction captured the approximate magnitude of the top three extremes in 1983, 1998 and 2017 (Figure 5). An analysis of flood reports from news

media and global disaster databases including EM-DAT and the Dartmouth Flood Observatory indicate that flooding along the Piura River occurred in each of these years, though not necessarily at the station itself.

## 4.4 Multi-model forecasts

For the multi-model forecast, least squares weighting results in a significantly higher weight (81%) assigned to the statistical model for Marañón, while the models are weighted equally (50% each) for Piura. In both cases, multi-model Pearson

correlation and RPSS values are similar to the independent statistical forecast model (Table 5). The Marañón multi-model detects all four true positives in the upper category – two more than GloFAS and the same as the statistical model. The Piura multi-model detects four true positives, one fewer than the statistical model and one more than GloFAS. For both Piura and Marañón, the multi-model forecast improves POD, FAR and TS compared to GloFAS (Table 6).

**Table 5:** Mean RPSS and Pearson correlation coefficients for each location and forecast approach.

| Site | Predictand | Statistical | | GloFAS | | Multi-model | |
|---|---|---|---|---|---|---|---|
| | | RPSS | Correlation | RPSS | Correlation | RPSS | Correlation |
| Marañón | MAM streamflow | 0.67 | 0.95 | 0.25 | 0.84 | 0.67 | 0.96 |
| Piura | FMA streamflow | 0.43 | 0.91 | 0.18 | 0.91 | 0.43 | 0.94 |

**Table 6:** Same as Table 5, but for POD, FAR and TS.

| Site | Predictand | Statistical | | | GloFAS | | | Multi-model | | |
|---|---|---|---|---|---|---|---|---|---|---|
| | | POD | FAR | TS | POD | FAR | TS | POD | FAR | TS |
| Marañón | MAM streamflow | 1 | 0.2 | 0.8 | 0.5 | 0.5 | 0.33 | 1 | 0.2 | 0.8 |
| Piura | FMA streamflow | 0.63 | 0.29 | 0.5 | 0.38 | 0.25 | 0.33 | 0.5 | 0 | 0.5 |

## 5 Discussion

### 5.1 Triggering early action

While verification metrics offer useful ways to evaluate forecast performance, a forecast's true value is determined by the end user (Hartmann et al., 2002). Because floods are the main hydro-meteorological threat in the Peruvian Amazon (IFRC, 2019) and Piura basins, correctly predicting the years with high seasonal streamflow are of outsized importance compared to predicting low-flow years. The Peruvian Red Cross early action protocol steps for flooding are triggered when a forecast

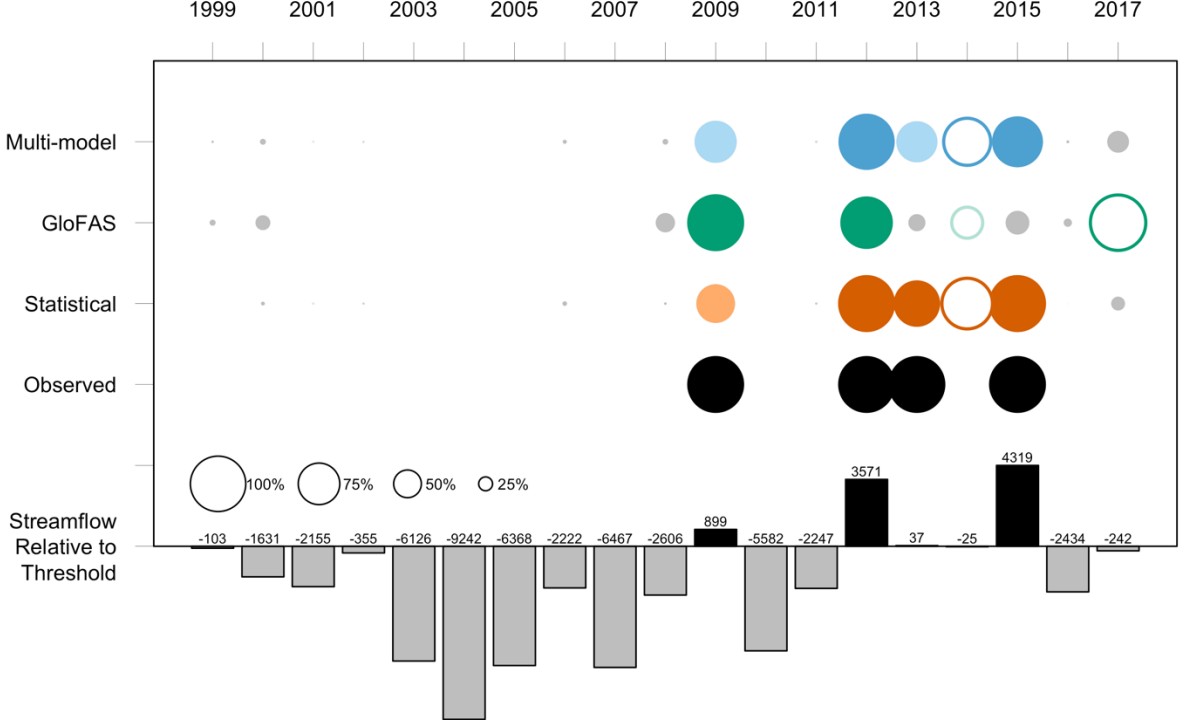

**Figure 6:** Marañón River at San Regis early actions triggered (≥75% probability of exceeding threshold) based on observed data (black) and season-ahead predictions from: statistical model (orange), GloFAS (green), and multi-model (blue). Dark colors represent a ≥75% probability of threshold exceedance; light colors represent a 50-75% probability of threshold exceedance; grey represents a <50% probability of threshold exceedance. Open circles represent false positives. Circle sizes are scaled to probability of threshold exceedance. Black (grey) bars indicate relative magnitude of streamflow compared to 80th percentile in m³/s.

predicts a 75% chance (probability) of streamflow above the 80th percentile (threshold). This criterion is applied to the three probabilistic forecasts (statistical model, GloFAS, and multi-model) to understand when actions would be triggered based on each forecast at San Regis on the Marañón River and at Puente Sánchez Cerro on the Piura River.

Based on the above criteria, four years in the historical record qualify for early action at San Regis (2009, 2012, 2013, 2015). Out of these four, the statistical model predicts action in three out of four years and GloFAS in two (2009 and 2012) (Figure

6). While an observed event does not necessitate observed flooding or flood impacts, the Centre for Research on the Epidemiology of Disasters (CRED) Emergency Events Database (EM-DAT) provides evidence of flooding in the western Amazon (Loreto region), though not necessarily on the Marañón, in 2012, 2013 and 2015 (the three highest seasonal averages on record) suggesting that early actions in these years could be warranted. In 2012 and 2015, when Marañón observed streamflow exceeds the threshold required for early action (26,671 m³/s) by over 3500 m³/s, the statistical model triggers with a 100% probability of threshold exceedance in both cases. In 2013, when observed streamflow is just 37 m³/s above the threshold, the statistical model predicts an 80.9% probability of threshold exceedance while the following year, when streamflow is 25 m³/s below the threshold, the statistical model predicts a 91.4% probability – its only false positive. GloFAS correctly triggers early action in 2009 and 2012 with 100% and 92% probabilities of threshold exceedance respectively while missing in 2013 and 2015 with predictions of 28% and 40% exceedance. In two out of the four years with observed triggers, the statistical model and GloFAS threshold exceedance probabilities differ by at least 50 percentage points (Figure 6). Additionally, in 2017, when streamflow misses the threshold for early action by only 242 m³/s, the two models differ in their predicted probability of threshold exceedance by 78 points. Collectively, these differences suggest that the two models capture distinct signals in years critical for disaster preparedness. Despite this, the multi-model least-squares ensemble forecast, weighted heavily toward the statistical model, mirrors the latter's predictions (Figure 6).

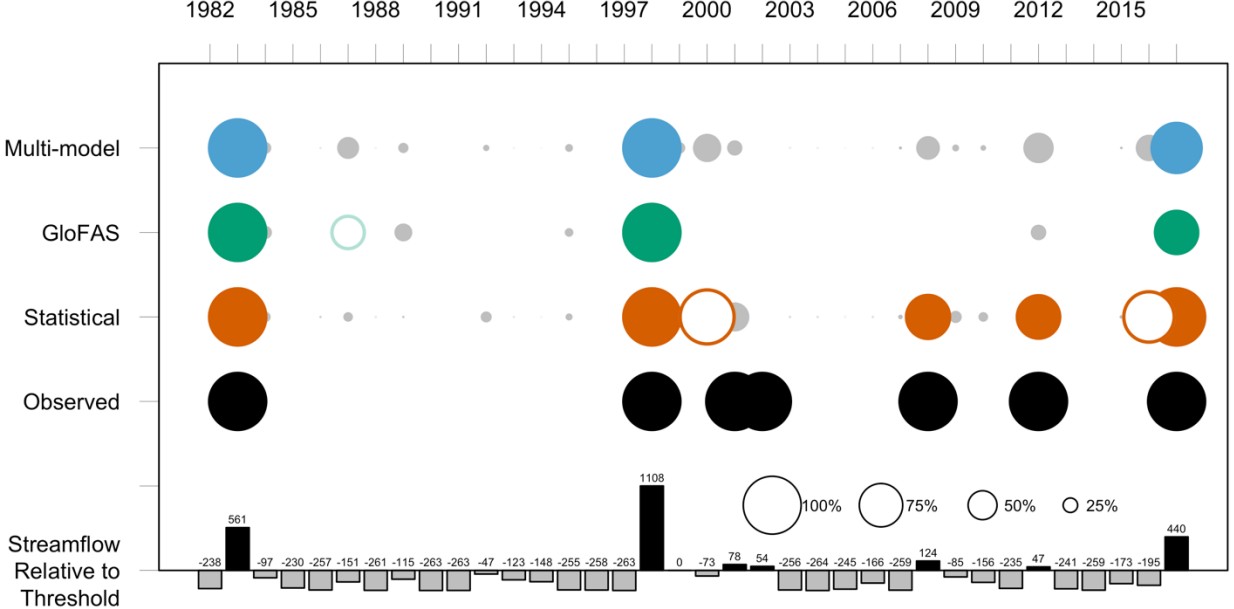

**Figure 7:** Same as Figure 6 for Piura River at Puente Sánchez Cerro.

At Puente Sánchez Cerro, all models trigger early actions during the three largest events in 1983, 1998 and 2017 – each of which resulted in significant impacts in the Piura River basin, collectively killing over 1000 people and affecting another 3.6

million (BBC News, 2017; Caviedes, 1984; EM-DAT, 1988; French and Mechler, 2017; USAID, 1998) (Figure 7). The statistical model includes two false positives in 2000 and 2016 with 93% and 87% predicted probabilities of exceedance (observed streamflow was at the 74th percentile in 2000). Additional historical years (2001, 2002, 2008 and 2012) also meet the criteria for early action with evidence of flooding in the Piura province, collectively resulting in 60 deaths and affecting 508,000 people (EM-DAT, 1988), although streamflow magnitudes were substantially lower. Of these the statistical model captured two (2008, 2012) while GloFAS failed to capture any.

A modified trigger mechanism captures some lower-magnitude events at San Regis; if early action is triggered based on just a 50% probability of exceeding the 80th percentile, the statistical model also triggers in 2009 and the multi-model triggers in 2009 and 2013 (thus each capturing all four observed events). However, caution is advised when reducing this threshold probability in practice as it will likely result in additional false positives. This study forgoes any systematic attempt to assess when early actions may or may not be warranted (e.g., determining an optimal threshold) in favor of illustrating that additional skill in detecting observed early action triggers is possible with the use of tailored statistical and multi-model forecasts. Further optimization of trigger probabilities may be possible and would require understanding regionally specific flood impacts and expected benefits of early action.

### 5.2 Varying the probability required to trigger action

Skill in detecting events is highly dependent on the threshold probability required to trigger early action. In general, a lower threshold for action will result in instances of worthy action but also more actions in vain. Conversely, a higher threshold for action will prevent false positives yet will reduce the likelihood that early actions will be taken when needed. This tolerance for false positives when implementing early action is an open question for decision makers and may depend on numerous technical, institutional and political factors outside the scope of this study. Here, the trigger mechanism for early action, which requires a 75% probability of streamflow above the 80th percentile, suggests a tolerance for a FAR of 0.25 for an unbiased forecast. Crucially, the small number of events when each forecast triggers early action (4 for San Regis and 7 for Puente Sánchez Cerro), creates significant uncertainty in the POD, FAR, and TS values calculated for the hindcast period (Figure 8). However, notwithstanding sources of model-related uncertainty, achieving an acceptably low FAR at the 75% probability level with 95% confidence is possible for Piura with the GloFAS and multi-model forecasts (Figure 8d), although no forecast achieves this for Marañón (Figure 8c). Importantly, uncertainty in these metrics is generally reduced in the statistical and multi-model forecasts compared to GloFAS (e.g., Figure 8a from 30% to 65% probability). The confidence intervals for the statistical and multi-model forecasts also tend to be offset in the more skillful direction compared to GloFAS. This is particularly the case for Threat Score (TS), a validation metric that describes the degree to which observed events correspond to forecast events, and is useful for evaluating the benefits of additional true positives against the costs of additional false positives when true positives are relatively rare (Figure 8e and 8f). However, there are notable exceptions to

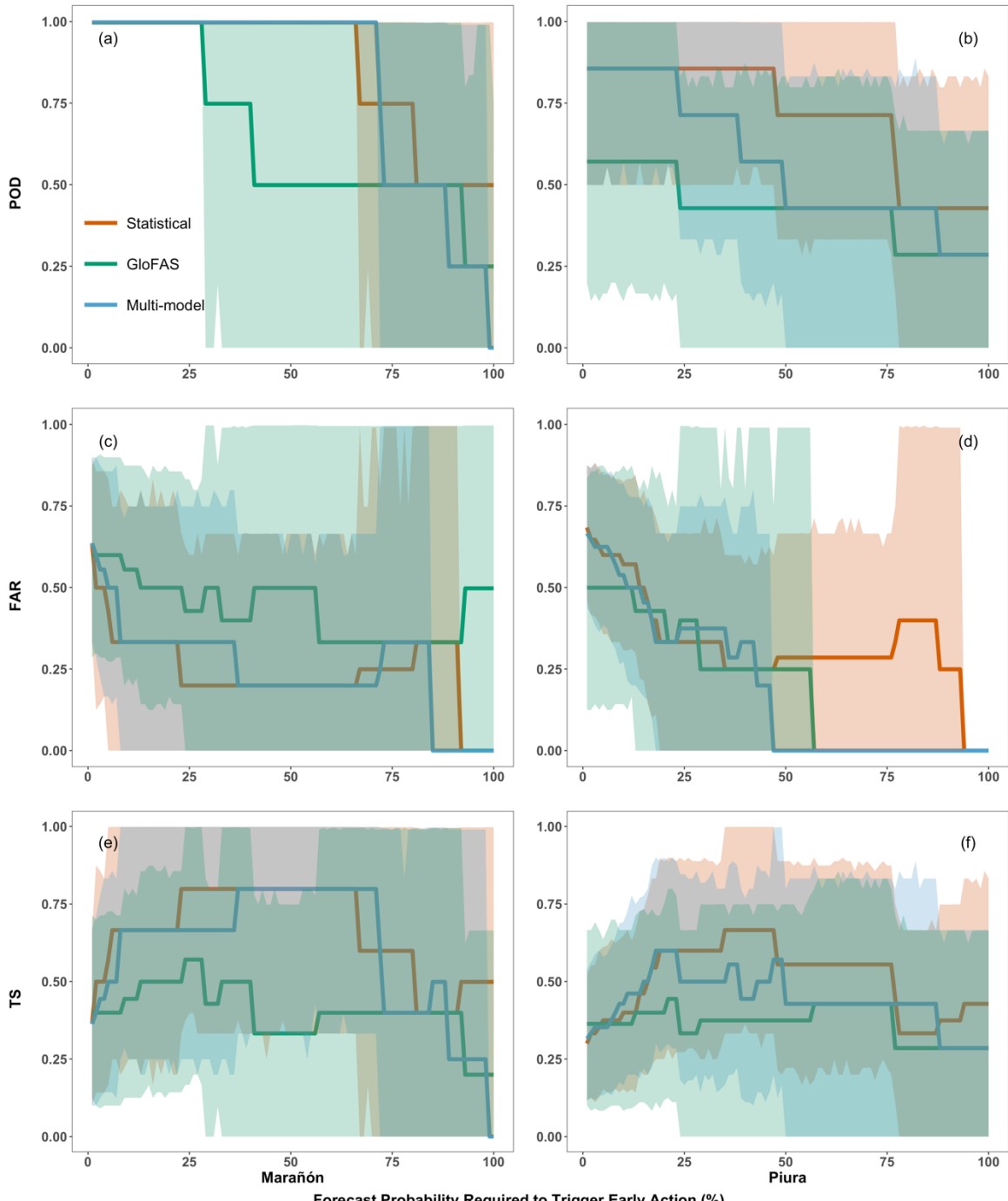


**Figure 8:** Probability of detection (POD), false alarm ratio (FAR) and threat score (TS) as a function of the threshold probability required to trigger early action for each location and forecast approach. Lines represent each metric calculated using hindcast data while ribbons represent sample size-associated uncertainty for each model at the 95% level, calculated via bootstrap resampling of the hindcast period (n=1000).

this trend, such as the large uncertainty in FAR for the statistical model at Piura above a 55% probability. While these results do not highlight an optimal probability threshold for decision makers, the statistical and multi-model forecasts generally appear more skillful across most probability levels. In addition, false positives incurred by reducing the trigger probability may also be offset by a stopping mechanism in which action is halted if the forecast is not confirmed 30 days later (IFRC, 2019).

**5.3 Implications of binary trigger mechanism**

The binary nature of the trigger mechanism is vulnerable to situations where similar observed conditions result in early action in one instance but not in another. Marañón River streamflow, which averages 24,600 m³/s during the MAM season, exceeded the 80th percentile by substantial margins in 2012 and 2015 (3,571 m³/s and 4,319 m³/s respectively), while in 2009 and 2013 it exceeded the 80th percentile by just 899 m³/s and 37 m³/s, respectively (Figs. 4 and 6). On the other hand, in

2014, streamflow averaged just 25 m³/s (0.09%) below the 80th percentile – warranting no early action based on the trigger criteria. Similar effects are visible in Figs. 5 and 7 for the Piura River: in 1999, streamflow was exactly equal to the 80th percentile and so did not count as an observed trigger (the stated mechanism requires that streamflow *exceed* the 80th percentile). It is also possible that observational error in streamflow measurements exceeds these differences. From an operational standpoint, such edge cases beg the question: should some amount of early action still occur? An observed

seasonal mean near the early action threshold, especially at the more variable Piura River, may contain much larger instantaneous discharge values and thus true flood risk may be obscured. Operationally, a trigger mechanism for early action at the Piura River should account for increased with-season variability of flows, perhaps by lowering the action threshold. Aside from these issues, a sharply defined threshold allows a potentially improper distinction between "worthy actions" and "actions in vain." In practice, absent a physical basis underpinning the action threshold, the difference in benefits resultant

from early action may be negligible for instantaneous discharge just above and below the threshold. This reinforces the need to also evaluate forecasts with complementary performance measures paired with local contextual knowledge. A modified trigger approach could incorporate multiple tiers of early actions triggered by increasing levels of forecast confidence. Likewise, if forecast confidence later decreases, a tiered stopping mechanism could halt actions in reverse order.

**6 Conclusion**

This paper describes a method by which locally tailored season-ahead statistical forecasts can improve the detection of trigger-based early actions and is illustrated with a case study for two sites in Peru. The statistical forecast developed in this study – as well as a multi-model ensemble forecast composed of the statistical and an operational physically-based model – consistently outperform the aforementioned physically-based model for both study locations. This method may be transferrable to other regions with evidence of seasonal streamflow predictability, especially in cases exhibiting a nonlinear

relationship between streamflow and climate variables. However, validation of NMME forecasts in other regions is advised

due to spatial variability in predictability. Opportunities for improving FbA via this framework may also be present in regions where global flood models are uncalibrated or display low skill.

While higher seasonal average streamflow values typically imply a greater probability of both flooding and the need for early action, lower seasonal average streamflow values may obscure high daily peaks that nonetheless result in flood impacts. Thus, even a perfect seasonal forecast may not reflect all instances where early action is justified. Additionally, because the statistical model developed here is optimized for performance across all years, further refinement prioritizing the detection of appropriate trigger levels for early action in high flow years may be warranted. Such efforts could involve alternative statistical or physical modeling frameworks, along with development of additional predictors and evaluation of category selection applied in the prediction process. Future work could also consider machine learning techniques with the goal of leveraging remotely sensed data to detect antecedent conditions at a subbasin scale and the state of the climate system.

*Code availability*. Code used in this study is available upon request.

*Data availability*. Streamflow data used in this study are from SENAMHI. While the dataset is not public, it may be made available upon request. PISCO precipitation data are available at piscoprec.github.io. Climate data obtained from NOAA are available at noaa.gov.

*Author contributions*. PB was responsible for conceptualization. CK developed and evaluated the prediction model with input from PB and DL. JB facilitated access to project resources (including datasets and documents) and provided contextual information. CK prepared the manuscript with editing contributions from all authors. PB and DL were responsible for project administration and PB was responsible for funding acquisition.

*Competing interests*. The authors declare that they have no conflict of interest.

*Acknowledgements*. Support for this research was provided by the Graduate School and the Office of the Vice Chancellor for Research and Graduate Education at the University of Wisconsin-Madison with funding from the Wisconsin Alumni Research Foundation (WARF). This research was also partially funded by a UW2020 grant supported by WARF. NMME project and data dissemination is supported by NOAA, NSF, NASA and DOE. We acknowledge the help of NCEP, IRI and NCAR personnel in creating, updating and maintaining the NMME archive. We acknowledge the agencies that support the NMME-Phase II system, and we thank the climate modeling groups (Environment Canada, NASA, NCAR, NOAA/GFDL, NOAA/NCEP, and University of Miami) for producing and making available their model output. NOAA/NCEP, NOAA/CTB, and NOAA/CPO jointly provided coordinating support and led development of the NMME-Phase II system.

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
