# Peer review of "Leveraging multi-model season-ahead streamflow forecasts to trigger advanced flood preparedness in Peru"

_Natural Hazards and Earth System Sciences, 2021_

## Author Comment (AC1)

**Reply to Reviewer #1**

**Title:** Leveraging multi-model season-ahead streamflow forecasts to trigger advanced flood preparedness in Peru

**Author(s):** Colin Keating, Donghoon Lee, Juan Bazo, Paul Block

**MS No.:** nhess-2021-25

The authors thank Reviewer #1 for the constructive comments and feedback on our manuscript. Our specific replies are denoted in blue color and revised manuscript text is denoted by italics.

**General comments**

The paper under review addresses an important topic within the scope of the journal, is generally well written and structured. Figures are visually appealing (especially Fig.6 & 7). Datasets used are adequate for the purpose of the study. Methods are rather traditional statistics (fairly old-fashioned), mainly a linear regression on principal components, but presumably also quite robust. No non-linear transformations, no unconventional predictors. The multi-model approach mentioned in the title is interesting. The chosen performance metrics for validation are also suitable. According to the authors, the developed model is an improvement to the current operational methods in Peru.

I suggest adding "in Peru" to the title of the manuscript – or any other spatial restriction the authors consider appropriate – as the method was only tested for two rivers in this specific country, and includes predictors that might not be suitable in other areas of the world (e.g. sea surface temperature for ENSO condition). If the authors want to claim that their method is in general better than operational practices worldwide, this claim would have to be substantiated by additional model runs in different places.

We thank the anonymous Reviewer for these comments which have led to further improvements in the quality of the manuscript. We agree with the Reviewer's suggestion to amend the title of the manuscript to include "in Peru," which now reads:

*Leveraging multi-model season-ahead streamflow forecasts to trigger advanced flood preparedness in Peru*

The authors made their code available to review via a GitLab repository, which is much appreciated! The provided R scripts are well readable (although not entirely in agreement with modern style guides, e.g. https://style.tidyverse.org/) and seem to cover all steps mentioned in the manuscript, from data preparation to model building and plotting. I did not try to run the code, as the raw data is not provided, but the scripts make the conducted research transparent.

We thank the Reviewer for this comment. We note that for additional readability, we have restyled all scripts according to tidyverse formatting rules.

**Specific comments**

About the manuscript, I request the following clarifications and modifications:

1. Please clearly define the term "season-ahead prediction". The term could be interpreted as predicting one season from the previous season, but I assume that the authors mean to predict one season from just before the start of that very season, as the 1-month-ahead streamflow appears to be included as predictor. Does the model only predict the maximum streamflow at some point during the season, or also a timing? 3 months is still quite an uncertain timeframe.

To clarify the term "season-ahead prediction," lines 105-108 now read:

> *In this paper, we use the term "season-ahead prediction" to describe forecasting the mean streamflow for an upcoming three-month season issued at the start of that season. For example, a season-ahead prediction of January-February-March streamflow would be issued on December 31ˢᵗ and represents a prediction of the average streamflow over the upcoming three months.*

2. In the introduction and discussion there should be an additional paragraph putting the used methods in context of what is state of the art in international scientific literature – not only in Peru. The last two sentences of the conclusion are: "*(…) because the statistical model developed here is optimized for performance across all years, further refinement prioritizing the detection of appropriate trigger levels for early action in high flow years may be warranted. Such efforts could involve alternative modeling frameworks (e.g. logistic regression), additional predictors, and evaluation of category selection applied in the prediction process.*" - But that is not enough and should appear earlier in the paper. Also, an additional paragraph about ensemble theory / multi-model studies would be adequate.

We agree with the Reviewer and have subsequently added additional paragraphs in the introduction section to detail the range of current statistical modeling approaches in the literature. We have also added a paragraph providing background on multi-model techniques. Combined, lines 81-100 now read:

> *A common traditional approach for statistical hydrologic modeling is multiple linear regression (MLR), which relates a predictand to the linear combination of several predictor variables (Moradkhani and Meier, 2010). For categorical streamflow forecasts, logistic regression (for two categories) or multiple logistic regression (for three or more categories) has been used successfully (e.g., Wei and Watkins, 2011). Because these methods are prone to multicollinearity due to the overlapping signals present in many hydroclimate variables, techniques such as principal component regression (PCR; a combination of principal component analysis and MLR) and partial least squares regression (e.g., Lala et al., 2020) are employed to address this challenge. More recently, machine learning techniques, adept at capturing nonlinear relationships between predictors and a predictand, have been successfully applied to hydroclimate forecasting, including artificial neural networks (Zealand et al., 1999), random forest classification (Ali et al.,*

*2020; Lala et al., 2020) and support-vector machines (Asefa et al., 2006; Shabri and Suhartono, 2012). There is also increasing recognition that hybrid approaches combining statistical and dynamical techniques can offer greater accuracy than even state-of-the-art dynamical models (Cohen et al., 2019).*

*Multi-model techniques have been developed based on the assumption that errors present in individual models may cancel out, thus providing a multi-model average with greater skill than any individual model, and to bound forecast uncertainty based on the spread of model predictions. Several methods of combining models include equal weighting, linear regression and Bayesian methods that assign weights according to the probability that the model in question has the highest skill (e.g., Gneiting and Raftery, 2005). In some cases, multi-model ensembles have been shown to significantly increase forecast skill over the best performing individual model (e.g., Regonda et al., 2006), while not in other cases. For example, Bohn et al. (2010) note only modest improvement when using a least-squares weighted multi-model.*

3. Data: The authors should make very clear for the reader which data was used to fit the statistical models, i.e. how many observations, where does the target variable (y) come from and how certain is it, what exactly are the explanatory variables and how have they been treated (scaling etc.). Most of that information is somewhere in the manuscript, but it is not as clear as it should be on first reading. Table 3 could be a good place to collect this information.

We thank the Reviewer for pointing our attention to this. Regarding the target variable and its certainty, we obtained this dataset from the Peruvian Meteorological Agency, El Servicio Nacional de Meteorología e Hidrología del Perú (SENAMHI), and they have conducted appropriate quality assurance. Lines 170-172 have been revised to:

*Daily streamflow data for each location (1999-2017 at San Regis, 1971-2017 at Puente Sánchez Cerro) was provided by the Peruvian Meteorological Agency, El Servicio Nacional de Meteorología e Hidrología del Perú (SENAMHI), who performed appropriate quality assurance.*

We have also clarified where the target variable comes from and the treatment of explanatory variables (scaling to have unit variance but no transformations) at the beginning of Section 3.3. Lines 265-277 have been revised to:

*A principal component regression (PCR; coupled principal component analysis and multiple linear regression) framework is adopted to predict seasonal (3-month) average seasonal streamflow derived from daily streamflow observations obtained from SENAMHI as described in Sect. 2.5. The forecast for each location is composed of sub-models (multiple linear regression) composed of years in a particular climate state, as represented by the preseason (3-month average) value of MEI. This produces two sub-models for the Marañón River at San Regis and three for the Piura River at Puente Sánchez Cerro. A hindcast assessment is*

*conducted by evaluating each year in the historical record using the appropriate sub-model to predict seasonal streamflow. For example, in 1998, the preseason (NDJ) average MEI value is 2.43, thus the positive phase sub-model is selected to predict Piura River FMA streamflow. Predictor variable types listed in Table 2 may be included in some sub-models and not others, subject to their correlation with streamflow in that phase (Table 3). To be included, the predictor in question must be both significantly correlated with streamflow across all years and significantly correlated with streamflow in the subset of phase-specific years. A principal component analysis is conducted on eligible predictors which are first scaled to have a unit variance. A subset of PCs is retained according to North's Rule-of-Thumb (North et al., 1982) for input into the multiple linear regression.*

We have also revised Table 3 (reproduced below) to include the number of observations (years) for each sub-model and the subset of predictors retained for each sub-model.

**Table 3:** Final predictors included in each sub-model.

| Site | Sub-model | Number of observations | Predictors retained from Table 2 | PCs retained | PC1 % variance explained | PC2 % variance explained |
|------|-----------|------------------------|----------------------------------|--------------|--------------------------|--------------------------|
| Marañón | Negative Phase | 12 | SST, SLP, SF, SM | 1 | 61 | 22 |
| | Positive Phase | 7 | SST, SLP, SF, SM, P | 1 | 87 | 9 |
| Piura | Negative Phase | 11 | SST, SLP, SM, P(GCM) | 1 | 74 | 15 |
| | Positive Phase | 14 | SST, SLP, SF, SM, P, P(GCM) | 1 | 78 | 13 |
| | Neutral Phase | 11 | SST, SLP, SM, P, P(GCM) | 1 | 68 | 15 |

4. "*There are numerous methods for selecting the appropriate number of PCs to retain; here, the first two PCs are retained unless the model has two or fewer predictors, and then only the first PC is retained.*" (254-256). How is the selection of only 2 PCs motivated? Contributions may differ during the seasons or per region, but at least some sort of check should be presented, e.g. by plotting the cumulative explained variance for El Niño and La Niña (or any other method the authors prefer to make this point that 2 PCs are sufficient). According to Table 3, only in one case have there been 2 PCs used – in all other cases only 1, so it is only linear regression with 1 predictor? Or 1 PC plus the streamflow before the start of the season?

Thank you for this comment; we have revised the method by which we retain PCs in our model and have revised Table 3 (reproduced above) to include the percent variance explained by the first and second PC. The revised process by which PCs are retained for each phase's sub-model are described on lines 275-281 as follows:

> *A principal component analysis is conducted on eligible predictors which are first scaled to have a unit variance. A subset of PCs is retained according to North's*

*Rule-of-Thumb (North et al., 1982) for input into the multiple linear regression, given as:*

$$y_t = \beta_0 + \beta_1 x_{1,t} + \cdots + \beta_n x_{n,t} + e , \qquad (1)$$

*where $y_t$ is observed seasonal streamflow in year t, $\beta_0$ is a constant, $\beta_1 \ldots \beta_n$ are regression coefficients, $x_{1,t} \ldots x_{n,t}$ are the PCs retained, and e is the residual or error. If North's Rule-of-Thumb indicates that no PCs are non-overlapping then only the first PC is retained.*

5. A critical point, acknowledged by the authors, is the selection of a threshold to issue an emergency. In my opinion this problem could be communicated better to the decision makers if a full probability distribution of expected streamflow were predicted, rather than a point estimate. Bayesian regression would be the adequate tool, then. As the statistical model presented by the authors appears to be very simple (linear regression with 1 or 2 predictors), implementing this in a Bayesian framework should be feasible. In that case, also Bayesian decision theory could be applied for the threshold selection. Apparently the authors create an error distribution by sampling the model residuals 1000x with replacement, which might end up in similar estimates, although with slightly different interpretation. At least the authors should discuss the probabilistic output in more detail, and also discuss how this probabilistic output can be used in risk communication and decision making for the problem at hand.

We thank the Reviewer for this comment and acknowledge the potential value of alternative modeling approaches (e.g. Bayesian regression/inference), especially for threshold selection. We emphasize that this study illustrates the potential for tailored statistical approaches to complement operational physical forecasts, and acknowledge that a range of alternative statistical approaches may offer enhanced skill. We agree these alternative approaches warrant consideration in future work, especially for developing specified guidance for stakeholders.

As mentioned by the Reviewer, we undertake a simplified ensemble generation process to create a probabilistic forecast distribution. Again, alternative approaches are available, however we are not focused on selecting the 'best' approach, particularly since that clearly differs by case study, disaster, region, etc. However the Reviewer's point regarding the need to better emphasize the importance of the probabilistic output in our methods is well received. We have revised lines 281-282 as follows:

> *The creation of probabilistic forecasts are essential as early action decisions are conditioned on the forecast likelihood of an extreme event exceeding the $80^{th}$ percentile.*

We have also added more detail about the probabilistic output to lines 421-432 which now read:

> *The primary focus of this study is to predict the occurrence of high flow conditions to initiate flood preparedness actions, based on a sufficient percentage of the probabilistic prediction surpassing a pre-defined threshold. The probabilistic statistical forecast model at each location effectively captures interannual*

*variability and extremes (Figs. 4 and 5). For the two most extreme years in the observed record (2012 and 2015 for Marañón; 1983 and 1998 for Piura), the full distribution of predicted streamflow falls above the 80th percentile of observed streamflow (black dashed line). In these years, decision-makers are highly certain of an impending extreme event. However, for the majority of years, some smaller fraction of the forecast distribution falls above the 80th percentile threshold, presenting a greater challenge (less certainty) in decision making.*

We agree with the Reviewer that utilizing probabilistic outputs is important in risk communication and decision making. We specifically address these issues in Sections 5.1 and 5.2, however we also acknowledge that there is room for improvement with respect to integrating probabilistic forecast output into decision making. This may include optimizing trigger thresholds, the probability required to surpass this trigger to initiate action, and exploration of the tradeoff in forecast skill and increased lead time for actions available at a range of lead times, all in the context of stakeholder tolerance for false positives and expected benefits. Indeed this is an active line of research in our group, however moves beyond the scope of this paper.

6. The multi-model seems to be dominated by the linear regression model. If this is the case, the authors could discuss which other models might be suitable to include in future multi-model ensembles.

We agree with the Reviewer's observation that the multi-model is dominated by the statistical model (we note that this is now the case for only one site in our revised analysis with an updated PC retention and predictor selection method). Ideally, members of a multi-model should each contribute skillfully such that errors in any single model are balanced by the other models. In our case, the global physical model (GloFAS; currently used for early warning decision-making) lacks sufficient skill at our study sites (for the lead time evaluated) to improve upon the statistical model or counter-act its errors. A calibrated basin-scale physical model may be better suited and more skillful than the bias-corrected GloFAS forecast when coupled to one or more GCMs with demonstrated predictive skill in the region (e.g. NCEP CFSv2 and NASA GEOS-S2S for coastal northern Peru, according to the work of our colleagues in atmospheric science). However, given that our statistical model at Piura is already forced using GCM precipitation predictions, it is not clear that additional skill would be realized in a multi-model. A challenge of modeling at our present study sites is data scarcity; however, machine learning techniques that leverage remotely sensed data (e.g. detecting antecedent soil moisture conditions or the state or direction of the atmospheric-oceanic system) could potentially offer avenues for improvement. To that end, we have added the following text to our conclusion (lines 569-571):

> *Future work could also consider machine learning techniques with the goal of leveraging remotely sensed data to detect antecedent conditions at a subbasin scale and the state of the climate system.*

We also note that the relative skill of the statistical and physical models (and thus weighting in the multi-model) may also be dependent on lead time, seasonality, and antecedent conditions. For example, the global dynamical model may be relatively more skillful at shorter lead times due to

its ability to include the effects of recent precipitation. At our study site, skill at shorter lead times may inform early actions relevant to the disaster event.

In conversations with our colleagues in the social sciences, we have learned that stakeholder buy-in – a critical step for creating forecasts that add value – may be easier to achieve with a simple model compared to a model that is more opaque or complicated. Further, the simple statistical model presented here performs quite well overall, and while a more complex model may perform marginally better, the overall gains are likely minimal compared to efforts placed on proper forecast dissemination, communication or training of stakeholders, etc.

**Technical corrections**

1. All tables would benefit from some formatting.

We have re-formatted all tables to improve clarity and readability.

2. In Table 2, the letters J and F are used without explanation. I assume it is January and February, respectively, as the authors write in the text that the high streamflow seasons in the basins are FMA and MAM, respectively. January and February would therefore correspond to a 1-month-ahead value. However, that should be stated explicitly in the text and above the table – or more clear abbreviations like "Jan" and "Feb" should be used.

We have added the following clarifying text to the Table 2 caption:

*"J (F) indicates January (February)."*

3. Especially the very important "predictors" column in Table 3 consists of abbreviations with distracting line breaks. As the columns of that table are repeated, consider arranging the "negative phase" "positive phase" and "neutral phase" in rows rather than columns, and use the free space to add more columns giving detailed information on the models, like the number of observations, PC2 explained variance, maybe even the cross-validation score. Consider removing the bold rectangle and make the font of the column/row names bold instead.

We thank the Reviewer for these helpful suggestions. We have revised Table 3 (reproduced above), switching rows and columns and adding two additional columns for PC2 variance explained and number of observations.

4. There is a LICENSE file included in the GitLab repository, but no README and CITATION files. I would like to encourage the authors to add these two missing components, although it is not a criterion for acceptance of the manuscript.

We thank the Reviewer for this suggestion and have added README and CITATION files to the GitLab repository.

---

## Author Comment (AC2)

**Reply to Reviewer #2**

**Title:** Leveraging multi-model season-ahead streamflow forecasts to trigger advanced flood preparedness in Peru

**Author(s):** Colin Keating, Donghoon Lee, Juan Bazo, Paul Block

**MS No.:** nhess-2021-25

The authors would like to thank Reviewer #2 for the constructive comments and feedback on our manuscript. Our specific replies are denoted in blue color and revised manuscript text is denoted by italics.

This paper describes evaluation of seasonal flood forecasts over Peru. The results are a key part of developing forecast-based early warning systems, where a robust understanding of model skill is crucial. The paper addresses an important question, the results are interesting and the manuscript is well structured and clear.

I am happy to recommend publication, after the authors address a few comments, below.

1. False alarm ratio (FAR) is calculated by counting #false alarms and #triggers and dividing the former by the latter. This is fine and is the standard method to do so. However it does mean that the sample is relatively low (particularly for seasonal reforecasts), leading to high uncertainty on the values. e.g. The sample size at Maranon is 19, but there are many fewer flood events and triggers than this. It is possible to partially address with uncertainty ranges on the verification statistics, calculated through a standard bootstrap resampling method (i.e. pick a 19 years with replacement from Maranon, recalculate FAR/ HR/ POD, and repeat). I would like to see this uncertainty added to figure 8, and its implications discussed.

We thank the Reviewer for bringing up this important issue regarding the uncertainty associated with a small sample size. We appreciate this suggestion to add uncertainty ranges to Figure 8 and have revised the figure accordingly (reproduced below). Additionally, we have discussed the implications of this uncertainty in section 5.2. Lines 471-496 have been revised to:

> *Skill in detecting events is highly dependent on the threshold probability required to trigger early action. In general, a lower threshold for action will result in instances of worthy action but also more actions in vain. Conversely, a higher threshold for action will prevent false positives yet will reduce the likelihood that early actions will be taken when needed. This tolerance for false positives when implementing early action is an open question for decision makers and may depend on numerous technical, institutional and political factors outside the scope of this study. Here, the trigger mechanism for early action, which requires a 75% probability of streamflow above the $80^{th}$ percentile, suggests a tolerance for a FAR of 0.25 for an unbiased forecast. Crucially, the small number of events when each forecast triggers early action (4 for San Regis and 7 for Puente*

*Sánchez Cerro), creates significant uncertainty in the POD, FAR, and TS values calculated for the hindcast period (Figure 8). However, notwithstanding sources of model-related uncertainty, achieving an acceptably low FAR at the 75% probability level with 95% confidence is possible for Piura with the GloFAS and multi-model forecasts (Figure 8d), although no forecast achieves this for Marañón (Figure 8c). Importantly, uncertainty in these metrics is generally reduced in the statistical and multi-model forecasts compared to GloFAS (e.g., Figure 8a from 30% to 65% probability). The confidence intervals for the statistical and multi-model forecasts also tend to be offset in the more skillful direction compared to GloFAS This is particularly the case for Threat Score (TS), a validation metric that describes the degree to which observed events correspond to forecast events, and is useful for evaluating the benefits of additional true positives against the costs of additional false positives when true positives are relatively rare (Figure 8e and 8f). However, there are notable exceptions to this trend, such as the large uncertainty in FAR for the statistical model at Piura above a 55% probability. While these results do not highlight an optimal probability threshold for decision makers, the statistical and multi-model forecasts generally appear more skillful across most probability levels. In addition, false positives incurred by reducing the trigger probability may also be offset by a stopping mechanism in which action is halted if the forecast is not confirmed 30 days later (IFRC, 2019).*

[Figure]

**Figure 8:** Probability of detection (POD), false alarm ratio (FAR) and threat score (TS) as a function of the threshold probability required to trigger early action for each location and forecast approach. Ribbons represent sample size-associated uncertainty at the 95% level, as calculated via bootstrap resampling of the hindcast period (n=1000).

2. In addition to point 1 above. The sample size means there is an inevitable an aspect of forecast behaviour which is not captured in the reforecast - and bootstrap resampling across years is unable to quantify this. To explain with an example: in 2013 the statistical model predicts 94% probability of exceedence, which is observed. In the evaluation this year will always turn up as a 'hit' for any threshold less than 94%. However if we take the probability as a reliable representation of likelihood, there is still a chance that it would have been a false alarm (i.e. 6%). Similarly, there is a chance that every probability which resulted in a trigger was a false alarm (as long as that probability wasn't 100%). This is an unavoidable result of small sample size - and one which bootstrapping will not quantify - so I am not suggesting any change. However I suggest the authors reconsider their conclusion "L483 Detection of additional high flow events is possible by lowering the forecast probability ... while maintaining a low false alarm ratio". This is only true for this particular realisation of the reforecast. If you lower the probability threshold, there will always be more chance of false alarms when you trigger, by definition. You might get lucky, but then again you might not. It is important to be clear about this otherwise misleading conclusions may be reached, e.g. L427 suggests a lowering of the trigger to 50% may capture many more events, "without additional false positives". This is highly contingent on the particular realisation of the reforecast. A decision-maker may read this paper and decide to take action when the forecast probability is 50%, as they understand that this has an FAR of 0%. But, the chance that action on a forecast of exactly 50% will be in vain is ... 50% (assuming the probability is reliable). So there is a good chance they may be in for a nasty shock! I suggest the authors rethink their advice on lowering the trigger without consequence.

We thank the Reviewer for this comment and in consideration of this have removed the conclusion on line 483. We have also revised lines 461-465 to the following:

> *A modified trigger mechanism captures some lower-magnitude events at San Regis; if early action is triggered based on just a 50% probability of exceeding the 80th percentile, the statistical model also triggers in 2009 and the multi-model triggers in 2009 and 2013 (thus each capturing all four observed events). However, caution is advised when reducing this threshold probability in practice as it will likely result in additional false positives.*

3. The statistical model uses antecedent SST as a predictor (capturing ENSO activity). It also uses a precipitation forecast from the NMME. But what about using the SST forecast from the NMME? If ENSO state is a strong forcing of rainfall/streamflow, then I would imagine that the FMA SST is more strongly related to streamflow than DJF SST? Possibly the precipitation forecast may capture some of this future signal - although precipitation errors are well known. I hope that the authors might consider adding this, as it may increase the skill even further and lead to a better early warning.

We thank the Reviewer for this suggestion on how we might further improve forecast skill. We had initially explored the use of NMME forecasts of SST as potential predictors but found that this yielded no improvement in skill (in terms of correlation and RPSS) for predicting Piura streamflow. Specifically, we created a predictor from the average of the two NMME models identified in this paper (GEOSS2S and CFSv2), over the Niño1+2 region (80-90W; 0-10S) over the FMA season, issued Feb 1st. We selected SSTs in the Niño 1+2 region because correlation

between observed FMA Niño 1+2 anomaly and Piura streamflow is very strong (0.82, compared with the correlation between observed FMA average SST anomaly in the Niño 3.4 region and Piura streamflow, at 0.30). However, correlation between Piura streamflow and predicted Niño 1+2 SSTs (from the two NMME models) is 0.74, less than the correlation between Piura streamflow and predicted NMME precipitation (0.84).

Additionally, we do not observe any significant improvements in prediction skill for years critical for flood preparedness. Given this result, we have chosen to retain our original inclusion of NMME precipitation predictions here, although we acknowledge that the NMME SST forecast performs almost equally well for Piura.

4. Can you show the weightings for the statistical model? The results are shown from cross-validation leave-one-out (which is appropriate). But if you built the model again using all years, this would be useful to show the relative importance of each predictor.

We appreciate the Reviewer's interest in the statistical model weightings. During principal component regression, the set of predictors are transformed by PCA and a subset of the resulting principal components are retained for a multiple linear regression. Thus, the model coefficients are based upon the PCs (which contain information from multiple predictors) rather than the predictors themselves, which are highly correlated. One way to extract the relative importance of predictors is through assessing their individual correlation with streamflow, as presented in Table 2, reproduced below. From this perspective, it appears that pre-season SSTs and precipitation are most important for Piura, closely followed by antecedent streamflow, NMME precipitation forecast, and SLPs. For Marañón, pre-season SSTs, antecedent streamflow, and SLPs are relatively more critical, followed by soil moisture and precipitation.

| Potential Predictor | Abbreviation | Spatial Region | Time Frame | | Pearson Correlation with Streamflow | | | | |
|---|---|---|---|---|---|---|---|---|---|
| | | | Piura | Marañón | Piura | | | Marañón | |
| Streamflow | SF | - | J | F | 0.84* | | | 0.84* | |
| Precipitation | P | Basin-Avg | J | JF | 0.88* | | | 0.68* | |
| Soil Moisture | SM | 1st PC of statistically significant ($p < 0.05$) regions within 12N to 23S, 35W to 81.5W | J | F | 0.69* | | | 0.74* | |
| Air Temperature | T | Basin-Avg | J | F | 0.26 | | | 0.11 | |
| GCM Precipitation Forecast | P(GCM) | 4.5S to 5.5S, 79.5W to 80.5W | FMA | - | 0.84* | | | - | |
| | | | | | El Niño | Neutral | La Niña | El Niño | La Niña |
| Sea Surface Temperature | SST | 1st PC of NIPA-identified regions | NDJ | DJF | -0.79* | -0.90* | 0.85* | -0.93* | -0.80* |
| Sea Level Pressure | SLP | 1st PC of NIPA-identified regions | J | F | -0.82* | -0.74* | 0.79* | 0.90* | -0.72* |

Another way to indirectly assess the significance of each predictor would be to test the correlation strength between each predictor and the first PC of all predictors. Through some additional analysis outside this paper, we note that, for Piura, this first PC correlates most strongly with soil moisture in the negative phase (La Niña years) and precipitation in the neutral

and positive phases (neutral and El Niño years). For Marañón, the first PC is most highly correlated with SLP in the negative phase and precipitation in the positive phase.

5. The GloFAS seasonal forecasts are publicly available on the 10th of every month - not the first, as is stated in L274 (see https://www.globalfloods.eu/technical-information/glofas-seasonal/ - NB they are initialised on the 1st but there is a lag until they are available, which may be where the confusion arises). Does this change the potential for early action, as the first month is almost half over before the GloFAS forecast is available? There are a few possibilities:

- if the action is strictly constrained to the start of the month, the GloFAS run from the previous month is the only available run, so this should be used instead in the comparison

- if it is OK that no forecast is available until the 10th, then the statistical model could (in theory) include additional information on the streamflow/SST/precip in the first few days.

The authors may want to follow either (or neither) of these ideas. But at least please comment on this issue of forecast timeliness in the text.

Thank you for bringing this point to our attention. In an operational setting, according to the Peruvian Red Cross flood early action protocol, forecasts are issued on a rolling basis, with early actions taken any time streamflow forecasts are above the threshold. While for simplicity we issue our forecasts at a fixed date annually, it would be acceptable to issue the forecast on the 10th of the month (when GloFAS seasonal forecasts become available). We therefore opt to modify our statistical and multi-model forecast issue date accordingly. We have revised lines 105-110 to reflect this:

> *In this paper, we use the term "season-ahead prediction" to describe forecasting the mean streamflow for an upcoming three-month season issued at the start of that season. Ideally, a season-ahead prediction of January-February-March streamflow would be issued on December 31st and represents a prediction of the average streamflow over the upcoming three months. In practice, due to lags in data availability and for purposes of direct comparison with a physically-based model, forecasts developed in this paper are issued on the 10th day into the three-month season.*

We have also revised lines 196-199 to reflect this change:

> *Predictions of seasonal (three month) average streamflow (m³/s) are issued on the 10th day into the three-month high flow season identified in Sect. 2, leveraging predictors based on values in the preceding months. Practically, issuing the forecast ten days into the forecast season allows time for large-scale climate data to be made available online, while also fostering a more direct comparison with GloFAS as described in Sect. 3.4.*

Lastly we have also revised lines 301-302 accordingly:

> *GloFAS forecasts are initialized on the first day of every month and become publicly available on the 10th day of the month.*

We note that this revision has the additional advantage of allowing a buffer window for other large-scale climate data sources to be made available online, and thus may be a more realistic issue date from an operational standpoint. We also acknowledge that additional predictor data from the first few days of the month could in theory be used, which may provide some additional forecast skill. We opt not to pursue this path because we expect any additional skill to be marginal due to the length of the forecast season and the slowly evolving nature of SSTs – a key predictor. This choice allows a more direct comparison with GloFAS seasonal because it is also initialized on the 1st of the month.

Minor comments

L41 Was FbA originally applied to droughts? As far as I am aware it is only now being developed for drought/food insecurity. Please clarify.

We thank the Reviewer for this comment and would like to clarify that to the best of our knowledge FbA has only been applied to droughts more recently. Our prior confusion likely stemmed from a report by Cabot Venton et al. (2012) which modeled the costs of early response versus late response for drought in Kenya and Ethiopia but did not involve the implementation of a forecast based early action schema. We have updated lines 40-42:

> *While FbA was initially applied to acute and slowly evolving threats like tropical cyclones, more recent efforts have targeted hydrological threats including extreme rainfall and flooding (e.g., Gros et al., 2019).*

L69 There is a bit of a logical jump from the previous paragraph, consider adding a linking sentence.

Thank you for bringing this to our attention, we have revised lines 69-72 to:

> *Improvement in the skill of hydrologic models over the last several decades has aided the development of FbA systems for flooding. Among hydrologic models, those that are physically based (or dynamical) simulate physical processes such as infiltration and runoff to produce streamflow predictions and are often forced with climate predictions downscaled from general circulations models (GCMs) or numerical weather models.*

L111 Slightly long sentence, could be split for readability.

We have revised this sentence (now lines 138-139) to:

*In the Amazon basin, the influence of climate variables on flood risk remains understudied (Towner et al., 2020) as a result of the nonlinear relationship between precipitation and streamflow (Stephens et al., 2015).*

L160 What do the colours represent in Figure 1? Satellite image, topography? If the latter then it needs a colorbar.

The coloring in Figure 1 represents idealized land cover. (This map layer was obtained from https://www.naturalearthdata.com/downloads/10m-raster-data/10m-natural-earth-2/.) We have updated the Figure 1 caption on lines 165-166 to clarify this:

*Case study locations with catchment boundaries delimited in red. Shading represents idealized land cover. Made with Natural Earth (naturalearthdata.com).*

L200 Table 2: Piura has correlation of 0.84 between J streamflow and FMA streamflow. However in L148 it states that there is no significant monthly autocorrelation in Piura streamflow. This seems to be inconsistent.

We thank the Reviewer for this comment and would like to clarify that there is significant monthly autocorrelation in Piura streamflow. 174-177 in the revised manuscript now read:

*Monthly mean streamflow at Marañón exhibits a sinusoidal autocorrelation structure, with statistically significant autocorrelation at one- and two-month lags as well as at interannual timescales. In contrast, streamflow at Piura exhibits significant autocorrelation at up to a three month lag yet minimal autocorrelation at interannual timescales, indicating a greater degree of variability in successive years.*

L200 Table 2: Maranon GCM precipitation forecast is not included as a predictor, presumably because the correlation with MAM streamflow is not sufficiently high. I wonder: is this because (a) there is low correlation between seasonal rainfall and seasonal streamflow at Maranon or (b) the GCM precipitation forecast at Maranon is not particularly good? It would be good to include this information. If the answer is (b), then see point 3 above: it may be that SST is a more valuable predictor to take from the GCM forecast.

Our initial goal for the statistical model was to forecast streamflow using three main classes of observed, pre-season variables: large-scale climate, precipitation, and (antecedent) streamflow. We deviated from this approach by including the NMME forecast for in-season precipitation for Piura, largely motivated by the relatively small basin size; this characteristic results in flashy-type floods and relatively limited watershed memory as streamflow moves quickly through the basin. On the other hand, the Marañón watershed, at 362,000 km², is significantly larger and preseason precipitation, particularly in the upper parts of the basin, correlates well with streamflow (0.68), due to travel times on the order of weeks to months. While including an NMME precipitation of prediction did not improve model skill, we agree with the Reviewer that

including an SST forecast from NMME may further improve the skill of the statistical model. This would require further analysis, complicated by the fact that significant SST regions differ by phase as shown in Figure 3b). We suggest that these additions should for now remain an avenue for future work and stress that this paper's goal is to provide an illustration of how statistical forecasts may complement operational physical models for improved preparedness, which we believe the Marañón case achieves at present.

L226 I am unsure what " n.d." means in this context.

We have updated this citation to:

(NOAA, 2020)

L228 A 3-phase ENSO model is used at Piura, although a 2-phase model does not affect material performance. Given the favouring of parsimonious models (L257), why do you retain the 3-phase model?

We appreciate the Reviewer's question here. Aggregate model performance does not differ drastically, though is slightly improved in the 3-phase version (RPSS of 0.43 vs 0.39; correlation of 0.91 vs 0.88). One key reason for selecting the 3-phase model was its improved performance in key years for flood preparedness. For example, the two-phase version underpredicts 2017 streamflow by 35% compared to 12% in the 3-phase. Additionally, the 3-phase version reduces the spread of model residuals: on average, the standard deviation of residuals in the 2-phase model is 94.6 while the 3-phase lowers this to 79.8. We have thus rephrased lines 254-255 to better reflect our rationale for choosing the 3-phase model:

(While a two-phase model for Piura was also tested, the 3-phase model improves performance, including in years critical for disaster preparedness.)

L272 Requires some more info on GloFAS: what is the reforecast period, which model version used, has the model been calibrated for these basins (where streamflow data has been shared with the GloFAS team, the model has been calibrated).

We have updated lines 299-301 to incorporate information on the reforecast period, model version and calibration:

Monthly hindcasts over the period 1981-2017 from the physically based GloFAS Seasonal model (version 2.0) for the two study locations are available from ECMWF (https://www.globalfloods.eu/general-information/data-and-services/). Both study locations were used for model calibration (E. Zsoter, personal communication, May 6, 2021).

L315 It would be useful to explicitly note how many upper tercile events are present for each site.

We have included this suggestion by amending lines 342-344 to:

*As previously stated, the extreme category is classified as seasonal streamflow values in the top 20% (80th percentile) of observations – four events for Marañón and seven events for Piura.*

L399 What is meant by 'observed trigger'? From the context I think it should read 'event'? 'Trigger' only applies in context of the forecast, not the observations (similarly used in L448).

We agree with this suggestion and have updated "observed trigger" to "event" on L399 and L448.

L450 I am not sure what is meant by "TS is maximised".

We have revised this section (see Comment 1) and have eliminated this phrasing.

L463 Another thing to consider with these close-to-threshold events is that the difference in streamflow between may very well be within the margin of observational error - particularly if the seasonal average is based on daily data (i.e. an accumulation of systematic/random errors over 90 days).

We agree and have revised line 495 to reflect this possibility:

*It is also possible that observational error in streamflow measurements exceeds these differences.*

L468 "two events of similar magnitude...are likely to produce similar impacts with early actions likely to yield similar benefits". I am not sure it is reasonable to say this. Two seasons with similar average seasonal streamflow may have highly different subseasonal variability. For instance season A: all season just below the overtopping level without breaching, season B: a little way below season A average for the first month, but then increasing and repeatedly flooding in the next two months. A & B may have very similar average streamflow - but very different impacts.

We agree with the Reviewer's logic here and clarify that our intent was to illustrate the likely impacts due to instantaneous streamflow values. Therefore, we have revised lines 495-502 to the following:

*From an operational standpoint, such edge cases beg the question: should some amount of early action still occur? An observed seasonal mean near the early action threshold, especially at the more variable Piura River, may contain much larger instantaneous discharge values and thus true flood risk may be obscured. Operationally, a trigger mechanism for early action at the Piura River should account for increased with-season variability of flows, perhaps by lowering the action threshold. Aside from these issues, a sharply defined threshold allows a*

*potentially improper distinction between "worthy actions" and "actions in vain." In practice, absent a physical basis underpinning the action threshold, the difference in benefits resultant from early action may be negligible for instantaneous discharge just above and below the threshold.*

---

## Author Response (AR2)

**Reply to Reviewer #1 (Report #2)**

**Title:** Leveraging multi-model season-ahead streamflow forecasts to trigger advanced flood preparedness in Peru

**Author(s):** Colin Keating, Donghoon Lee, Juan Bazo, Paul Block

**MS No.:** nhess-2021-25

The authors would like to again thank Reviewer #1 for the feedback on our revised manuscript. Our specific replies are denoted in blue color and revised manuscript text is denoted by italics.

1. With all the additions from the previous revision round, the paper is a little lengthy now and the chapters could be separated better. Especially the chapters 3.1 and 3.2 repeat content that is visible in the tables, and describe things that have not been used (e.g. l. 264 "Selecting SST regions based on the preseason state of the Niño 1+2 anomaly index instead of MEI did not materially change results at Piura", l. 305–307 "A quantile mapping approach (…) did not substantially differ (…)". Slightly cutting these unnecessary parts should be enough.

We agree with the Reviewer that the manuscript has increased in length, and have accordingly cut the sentences mentioned above as well as additional text to remove information repeated in tables and elsewhere in the manuscript.

2. Please either merge "Results" and "Discussion" to "Results & Discussion" or separate more strictly. The Discussion chapter should give an honest evaluation of the overall results, and put them into context by citing relevant literature. In my opinion, chapter 4.2 is Methods rather than Results, while 4.1 contains Discussion parts (e.g. comparison to Bazo et al. in l. 354). The Discussion then presents 3 more figures. Some repetitions could be avoided from merging the sections.

We have merged chapters 4 and 5 into "Results and Discussion" as suggested by the Reviewer and have moved section 4.2 to the methods section.

3. I doubt that the term "principal component regression PCR" is adequate to describe your method. In my understanding, the term PCR suggests that a regression is applied in principal component space, by which I mean that all variables have been included in the PCA, and the resulting regression is then transformed back into the original feature space. You are using only 1 PC of selected variables and most other predictors are regular variables. I would rather write of linear regression and a PC predictor component. Also I find it a bit pointless to always stress the "multiple" linear regression, as most people doing linear regression use multiple predictors. It's ok if you just use your abbreviation MLR. The formulation in l. 271 "coupled principal component analysis and multiple linear regression" is wordy.

We have revised several instances of "multiple linear regression" to "MLR." Additionally, we have clarified our statistical method by revising lines 267-281 to

*The statistical forecast is composed of sub-models built only on data from years in a particular climate state, as represented by the preseason (3-month average) value of MEI. This produces two sub-models for the Marañón River at San Regis and three for the Piura River at Puente Sánchez Cerro. Each sub-model leverages a principal component regression (PCR) framework to predict seasonal (3-month) average streamflow derived from daily observations obtained from SENAMHI as described in Sect. 2.5. In this framework, a principal component analysis is conducted on eligible predictors (Table 2) which are first scaled to have a unit variance. A subset of PCs is retained according to North's Rule-of-Thumb (North et al., 1982) for input into a MLR model, however in all cases just one PC is retained, yielding a linear model of the form:*

$$y_t = \beta_0 + \beta_1 PC_1 + e \,, \tag{1}$$

*where $y_t$ is observed seasonal streamflow in year t, $\beta_0$ is the intercept, $\beta_1$ is a fitted regression coefficient, and e is the residual or error. Predictors may be eligible for inclusion in some sub-models and not others, subject to their correlation with streamflow in that phase (Table 3). To be included, the predictor in question must be both significantly correlated with streamflow across all years and significantly correlated with streamflow in the subset of phase-specific years. A hindcast assessment is conducted by evaluating each year in the historical record using the appropriate sub-model to predict seasonal streamflow. For example, in 1998, the preseason (NDJ) average MEI value is 2.43, thus the positive phase sub-model is selected to predict Piura River FMA streamflow.*

We prefer to keep the naming convention of PCR because we believe it aligns with prior literature's use of the term (e.g., Delorit and Block, 2017; Lins, 1985; Mortensen et al., 2018)

4. Please introduce abbreviations at the first occurrence of the term, and then always use the abbreviation afterwards. "Multiple linear regression (MLR)", "Threat Score (TS)", and others.

We thank the Reviewer for catching this and have abbreviated to MLR (l. 268, 270 and 279); EAP (l. 414); TS (l. 476, 480); POD (l. 480); FAR (l. 480).

5. Table 5 and Table 6 could be merged when arranging in rows rather than columns, similar to Table 3 (which looks good now). When doing so, it is much easier to visually compare the different models by all metrics. Consider to highlight the best score per metric in bold font.

We have merged Tables 5 and 6 (reproduced below) and have highlighted the best score per metric and site in bold. We agree this is a preferred illustrative approach.

**Table 5:** Mean RPSS, Pearson correlation coefficient, POD, FAR and TS for each location and forecast approach. Bold text indicates best score metric per site (ties between two models are both bolded).

| | Statistical | | GloFAS | | Multi-model | |
|---|---|---|---|---|---|---|
| | Piura | Marañón | Piura | Marañón | Piura | Marañón |
| RPSS | **0.43** | **0.67** | 0.18 | 0.25 | **0.43** | **0.67** |
| Correlation | 0.91 | 0.95 | 0.91 | 0.84 | **0.94** | **0.96** |
| POD | **0.63** | **1** | 0.38 | 0.5 | 0.5 | **1** |
| FAR | 0.29 | **0.2** | 0.25 | 0.5 | **0** | **0.2** |
| TS | 0.5 | **0.8** | 0.33 | 0.33 | 0.5 | **0.8** |

6. l. 46-54 The thematic jump from exposure/vulnerability to high temperatures in London requires rephrasing. As the rest of the article is about floods, the sentence should start with something like "In the context of heatwaves in London, (…)"

We agree and have revised lines 49-52 to

> *In the context of heatwaves in London, actions to reduce vulnerability for high-risk groups, such as ensuring indoor temperatures are below 26°C, are triggered when a forecast indicates temperatures of at least 32°C during the day and at least 18°C at night (Public Health London, 2018).*

7. l. 65 needs a comma. In addition, consider to split the long sentence after "protocols" (l.66)

Lines 64-68 have been revised to

> *In addition to short term weather forecasts, which are commonly viewed as skillful, medium to long range climate forecasts have also been demonstrated to improve preparedness protocols, resulting in reduced mortality, morbidity, and resource demands (Braman et al., 2013). However, their applications have been limited predominantly as a result of moderate forecast performance and significant uncertainty.*

8. l. 94 the assumption here is actually that the errors in individual models are uncorrelated. Correlated errors would not cancel out.

We agree and have revised (see comment 9).

9. l. 95 full stop after "individual model". Please rephrase the subsequent sentence.

We have revised lines 95-97 to

> *Multi-model techniques have been developed based on the assumption that individual model errors are uncorrelated, in which case a multi-model average*

*could provide greater skill than any individual model. Options for combining models include equal weighting, linear regression, or Bayesian methods (e.g., Gneiting and Raftery, 2005).*

References

Delorit, J. and P. Block (2017). Evaluation of model-based seasonal streamflow and water allocation forecasts for the Elqui Valley, Chile. Hydrology and Earth System Sciences, 21, 4711-4725. doi.org/10.5194/hess-21-4711-2017.

Lins, H. F. (1985). Interannual streamflow variability in the United States based on principal components. Water Resour. Res., 21, 691–701.

Mortensen, E., Wu, S., Notaro, M., Vavrus, S., Montgomery, R., De Piérola, J., … Block, P. (2018). Regression-based season-ahead drought prediction for southern Peru conditioned on large-scale climate variables. Hydrology and Earth System Sciences, 22(1), 287–303. https://doi.org/10.5194/hess-22-287-2018.